


# Drivers governing the seasonality of new particle formation in the Arctic

Dominic Heslin-Rees[1,2], Peter Tunved[1,2], Diego Aliaga[1], Janne Lampilahti[3], Ilona Riipinen[1,2], Annica M. L. Ekman[4,2], Ki-Tae Park[5], Martina Mazzini[6], Stefania Gilardoni[7], Roseline Thakur[3], Kihong Park[8], Young Jun Yoon[9], Kitack Lee[10], Mikko Sipilä[3], Mauro Mazzola[11], Radovan Krejci[1,2]

[1]Department of Environmental Science (ACES), Stockholm, Sweden.
[2]Bolin Centre for Climate Research, Stockholm University, Stockholm, Sweden
[3]Institute for Atmospheric and Earth System Research (INAR), Helsinki, Finland.
[4]Department of Meteorology, Stockholm, Sweden.
[5]Hallym University, Chuncheon, Gangwon, Republic of Korea
[6]Institute of Atmospheric Science and Climate (CNR-ISAC), Bologna, Italy.
[7]Institute of Polar Sciences (CNR-ISP), Milan, Italy.
[8]School of Earth Sciences and Environmental Engineering, Gwangju Institute of Science and Technology, Buk-gu, Gwangju, Republic of Korea
[9]Korea Polar Research Institute, Yeonsu-gu, Incheon, Republic of Korea
[10]Division of Environmental Science and Engineering, Pohang University of Science and Technology, Pohang, Republic of Korea
[11]Institute of Polar Sciences, National Research Council (CNR-ISP), Bologna, Italy

*Correspondence to:* Dominic Heslin-Rees (dominic.heslin-rees@aces.su.se) and Radovan Krejci (radovan.krejci@aces.su.se)





## Abstract

New particle formation (NPF) is the phenomenon wherein gaseous precursors form critical clusters of barely a few nanometres in diameter, after which, under favourable conditions these particles can grow to climate-relevant sizes. Here we present measurements from 2022 to 2024 of particle and ion number size distributions from the Zeppelin Observatory (ZEP), an Arctic research station situated on the western edge of Svalbard. NPF events begin in April and continue occurring into November. The events at the start of the NPF season (i.e. April/May) are considerably stronger (i.e. a larger production of nucleation mode particles). The peaks in NPF strength coincide with peaks in the solar insolation experienced by arriving air masses. During the summer period NPF events occur on 20-40% of days each month, however, there is a consistent decline in June. We show that the combined influence of solar radiation and the surface area of pre-existing aerosols (i.e. condensation sink, CS) are strong predictors for the likelihood of NPF. We develop a simplified predictive model which matches the frequency of NPF events identified via the classification schemes used in this study. We show that NPF events occur during the polar night (i.e. when the Sun does not pass above horizon), and speculate that these events are linked to high altitude air masses. Furthermore, we detail the likely geographic origins of nucleation within the Arctic, as measured at ZEP. We show that NPF events are considerably more likely to originate from the marine regions towards the west of Svalbard, particularly the Greenland Sea which presented the greatest likelihood that arriving air masses from this marine region would be linked to an NPF day. We also remark on the proportion of the Aitken mode particles within the Arctic that could originate from NPF; we show that NPF events lead to an increase in the number of Aitken mode particles. We measure over 50 NPF events where the nucleation mode particles grew beyond 25nm, a diameter representing the minimum activation diameter for particles to act as cloud condensation nuclei. Overall, we present a concise picture of the lifecycle of nucleation mode particles in the Arctic, including the effect wet scavenging has in reducing the condensation sink, which in turn encourages NPF events to occur.




## 1. Introduction

New particle formation (NPF) describes the production of secondary aerosol particles from precursor gases to clusters of molecules a few nanometres in diameter. Newly formed particles belong to the nucleation mode (i.e. typically <10nm) and may grow through the continued coagulation and condensation of vapours into Aitken mode particles (i.e. tens of nanometres in diameter). The phenomenon of NPF has been observed globally (Kerminen et al., 2018), however, in each region the chemical characteristics and controlling source and sink processes leading to NPF may vary.

Arctic aerosol seasonality is governed by a combination of atmospheric general circulation, meteorology, and several source and sink processes including the direct emission of local primary particles, NPF, the long-range transport of anthropogenic emissions from lower latitudes, and the removal of particles through wet scavenging (Garrett et al., 2011; Schmale et al., 2022; Tunved et al., 2013; Willis et al., 2018). One distinctive feature of the annual cycle is the Arctic haze in late winter and early spring, which is characterised by a sustained increase in the number of accumulation mode aerosol (i.e. particles typically around 100nm) (e.g. Shaw (1995)). Reduced wet scavenging and increased transport efficiency from anthropogenic source regions at lower latitudes during this part of the year give rise to this phenomenon (Garrett et al., 2011). The Arctic summertime, by contrast, is less influenced by long-range transported pollutants, and the increased precipitation effectively reduces the once pronounced accumulation mode. Moreover, increased photochemistry and biological activity (e.g. phytoplankton blooms) provide a source of precursor gases encouraging NPF and the growth of newly formed particles (Tunved, Ström and Krejci, 2013; Croft et al., 2016; Price et al., 2023); as a result, the smaller Aitken and ultrafine aerosol particles increase in number.

Changes in aerosol size, abundance and chemical composition can perturb the radiative balance either indirectly by acting as cloud condensation nuclei (CCN), altering cloud formation, brightness, and lifetime or directly through their interaction with both short- and long-wave radiation (Carslaw, 2022). The Arctic region can at times be CCN-limited, which means that small changes in CCN concentrations could potentially impact cloud properties via cloud-mediated indirect aerosol effects, which in this and similar environments could result in surface warming due to enhanced long-wave cloud forcing (Garrett et al., 2004; Garrett and Zhao, 2006; Mauritsen et al., 2011). Thus, local-regional scale production of aerosols during the summertime and changes to this production mechanism may have significant consequences for the Arctic (Kecorius et al., 2019). Still however, the impact of newly formed particles on Arctic cloud properties remains poorly quantified.



Depending on certain environmental factors, for example the rate of particle nucleation, the production rate and volatility of condensable vapours, and the condensation and coagulation sinks, to mention a few, these newly formed particles can reach climate-relevant sizes; In the Arctic, studies have suggested that particles as small as 25nm can participate in cloud formation by acting as CCN (Leaitch et al., 2016; Karlsson et al., 2020, 2021; Pöhlker et al., 2021; Gramlich et al., 2023; Motos et al., 2023) thus affecting cloud properties and climate (i.e., by influencing the radiative balance). Motos et al. (2023) predicted the smallest particle activation diameters in the summertime to be 20nm, with a modal average of 60 nm, but suggested that the dominant summertime Aitken mode originating from NPF may stay in the interstitial (inactivated) phase. Other studies have indicated that a substantial fraction of CCN in the Arctic are produced via NPF (Gordon et al., 2017; Merikanto et al., 2009).

Air ions are an additional focus area of aerosol effects, as they can play a role in nucleation (Hirsikko et al., 2011; Kirkby et al., 2011) and can influence aerosol particles through their formation and growth mechanism. Both positive and negative ions both contribute to the production of newly formed particles in the Arctic, via ion-induced nucleation (see Beck et al., 2021 for negative ions). The overall contribution of ions to NPF is disputed as quantifying their impact on nucleation rates in ambient conditions remains difficult. However, studies suggest that the dynamics of sub-2nm clusters is dominated by neutral clusters (Hirsikko et al., 2011; Kulmala et al., 2013) . Ions can help stabilise clusters which form initially (Yu and Turco, 2001) , with most particles formed from ion-induced nucleation reaching neutrality after growing beyond 2.5nm (Wagner et al., 2017).

Observations of ultrafine particles (~3-20nm) in the Arctic (onboard the Swedish icebreaker Oden, 70°N-85°N) were reported by Covert et al. (1996) and Wiedensohler et al. (1996). At Zeppelin Observatory (ZEP), Ström et al. (2003) and Tunved et al. (2013) presented measurements of small (>10nm) aerosol particles showing an increased concentration and frequency of formation events during the Arctic summer. Tunved et al. (2013) suggested that the summertime, coinciding with increased photochemical production of precursor gases, and the removal of larger pre-existing aerosol, generates favourable conditions in which NPF can occur. Other Arctic sites have been the focus of research into NPF, including Villum Observatory (see Nguyen et al. (2016)) and Summit (see Ziemba et al. (2010)), both on Greenland, and Barrow Alaska (Kolesar et al., 2017a). Other studies, e.g. Brean et al. (2023) present NPF data from several locations, including Alert, Villum, Tiksi, ZEP, Gruvebadet (Ny-Ålesund), and Barrow; a commonality shared by all Arctic sites was the occurrence of NPF events during the summer, and also that events coincided with air masses generally influenced by marine regions, including at Barrow and Alert whereby a correlation was observed between biogenic methane suphonate (MSA-) and summertime particle number concentrations (Leaitch et al., 2013; Quinn et al., 2002). Beck et al. (2021)  through



a comparison of measurements from Ny-Ålesund and Villum, showed that different nucleation mechanisms can
also occur; at Villum iodic acid was found to be the primary driver of NPF events, whilst at Ny-Ålesund NPF was
driven by sulphuric acid (SA) and ammonia. There is no evidence that anthropogenic precursors contribute to NPF
at ZEP (Schmale and Baccarini, 2021). Instead, DMS can serve as a precursor gas contributing to NPF. Strong
correlations between measured concentrations of DMS and the exposure of air masses to chlorophyll a have been
presented (Lee et al., 2020), which suggest marine DMS emissions are an important source of nucleating and
condensing material.  Observed DMS concentrations vary with airmass origin, season and residence time over
different marine source regions. For example, it has been shown that air masses exposed more to the Greenland
Sea, compared with the Barents Sea bring about higher DMS concentrations (Park et al., 2018). In the Arctic, there
are low concentrations of nucleating agents as opposed to continental rural and urban locations (Karl et al., 2012;
Pirjola et al., 2000).  However, in spring and summer increased solar radiation, a lower total surface area of pre-
existing particles (i.e. reduced condensation sink, CS), and coinciding phytoplankton blooms are shown to give
rise to precursor gas concentrations sufficient to sustain ion-induced nucleation, and growth of NPF-induced
particles beyond 20nm (Beck et al., 2021).

In this study, we present two and a half years' worth of measurements of particle and ion number size distributions,
including three summer periods (April 2022 – October 2024), whereby continuous measurements of sub-3nm up
to ~850nm particles have been performed at the ZEP, Svalbard, providing unique information from the initial
stages of nucleation up to the higher end of the accumulation mode.   NPF events were identified, formation rates
were calculated, and subsequent growth was analysed. The seasonality and coinciding environmental parameters
encouraging NPF events and subsequent growth have been explored in detail.

## 2. Materials and methods

### 2.1 Measurement site

Zeppelin observatory (ZEP) (78.90°N, 11.88°E, 474 m a.s.l.) is situated on the ridge of Mount Zeppelin, 2 km
south from Ny-Ålesund research village on the western edge of the Norwegian Svalbard archipelago. ZEP
represents regional background conditions in the Atlantic sector of the Arctic, as it is largely unaffected by local
air pollution due to its location well above Ny-Ålesund and the prevailing wind patterns driven by Kongsfjorden's
geography. ZEP is one of the most developed atmospheric observational sites in the Arctic where a broad set of
parameters has been measured for decades. It is part of numerous regional and global monitoring networks,
including the Global Atmosphere Watch (WMO/GAW), Integrated Carbon Observation System (ICOS), The



Aerosol, Clouds, and Trace Gases Research Infrastructure (ACTRIS) and the co-operative programme for monitoring and evaluation of the long-range transmission of air pollutants in Europe (EMEP). For a detailed

description of the observatory, local meteorology and climatology, history and observational program see Platt et al. (2022).

**2.2 Instrumentation**

For this study, an array of instruments has been used to measure aerosol and ion size distributions, between 0.8 and 850 nm including a Neutral Cluster and Air Ion Spectrometer (NAIS5, Airel Ltd. Tartu, Estonia), a

Nanoparticle Scanning Mobility Particle Sizer (Nano SMPS, TSI), and a Differential Mobility Particle Sizer (DMPS) system. Particle and ion number size distributions and concentrations form the core data used in this study. Several other supporting data sets were used for data analysis and interpretation and are described below. This study covers the period from 2022-04-17 to 2024-10-04.

**2.2.1 Differential Mobility Particle Sizer (DMPS) system**

Particle number size distributions, from 5 to 850nm, were measured using a custom-made twin-DMPS system with a closed loop sheath circulation and composed of two Hauke-type Differential Mobility Analysers (DMA). DMPS-1, with a short (5.3 cm) DMA, measured aerosol size distribution from 5 to 57 nm using sample-to-sheath air ratio close to 1:10. DMPS-2, with medium length DMA (28 cm), measured size distribution from 20 to 850 nm using sample-to-sheath air ratio close to 1:5. The aerosol particles were counted using condensational particle

counters (CPCs); a TSI model 3010 was coupled to the DMPS-1 and a TSI model 3772 was coupled to the DMPS-2. The total aerosol concentration was measured at a frequency of 1 Hz with the two CPCs. Particles > 3 nm were measured using the Ultrafine CPC (UCPC) TSI model 3776 (cut-off of 2.5nm) and particles > 10 nm with the CPC TSI model 3010 (cut-off of 10nm). The two DMAs were run in a stepwise mode, providing two particle number size distribution cycles approximately every 27 minutes. The overlap between the size ranges measured by both

DMPSs was used to check consistency and to merge both size distributions into one. Additional information can be found in Karlsson et al. (2021). The DMPS measurements were corrected for diffusional losses using the Particle Loss Calculator (PLC) according to Von Der Weiden et al. (2009) (see Fig. S1 in the supplement for details).

**2.2.2 Neutral Cluster and Air Ion Spectrometer (NAIS)**

The NAIS was utilised to measure concentrations of ions (charged particles and cluster ions) of both polarities between 0.8 and 40 nm, and particles between 2.5 and 40 nm. The NAIS is an aerosol mobility spectrometer; when

the sample is left unmodified it detects naturally-charged ions and particles, and when the corona charger is used to charge the particle population it measures all particles including the uncharged (Manninen et al., 2011; Mirme et al., 2007; Mirme and Mirme, 2013). It consists of two mobility analyser columns, with a total of 25 electrometers per column. Its lower size thresholds and high temporal resolution make it highly suitable for observing the early stages of NPF. The NAIS provides very accurate size information, regardless of the choice of the inversion algorithm used (in this case, the 25-channel V14.1 inverter algorithm) (Wagner et al., 2016). The lowest detection limit for the NAIS in particle mode is set based on the diameter of the corona charger ions, and the inability of the electrical filter to remove all naturally charged particles (Manninen et al., 2011), however, there is a broad consensus that measurements of concentrations of particles below 2.5nm are inaccurate.

The NAIS at ZEP was installed with a very short metallic inlet, less than 1m long, to minimise diffusional losses. The inlet pointed slightly downward, and a metallic rain shield was attached at the end, to ensure that rain did not enter into the instrument. The NAIS inlet was heated to prevent ice blocking the inlet, and also to limit water condensing inside the tubing. The NAIS was serviced and cleaned once a year. Additional aspects of the NAIS observations, including the apparent overestimation of particle number concentrations when compared to DMPS measurements (see also Kangasluoma et al., 2020) and the loss of negative cluster ions can be found in section S2.6 of the supplementary material.

### 2.2.3 Complementary observations

A Nanoparticle Scanning Mobility Particle Sizer (Nano SMPS, TSI) comprising of a nano-differential mobility analyser (nano-DMA, TSI 3085, USA) and an ultrafine CPC (TSI 3776, USA), provided size distributions of nanoparticles (3-60nm). See Lee et al. (2020) for further details. Atmospheric Dimethyl Sulphide (DMS) was measured using an analytical system in which DMS is trapped and eluted, before being quantified using gas chromatography equipped with a pulsed-flame photometric detector. The detection limit is approximately 1.5 pptv given air samples of around 6 litres. For more details, see Jang et al. (2016) and Park et al. (2018). Meteorological parameters including visibility, relative humidity, wind speed and direction, ambient temperature and pressure were also further utilised throughout the study.

### 2.3 Data Analysis of aerosol measurements

For processing of the NAIS data the *nais-processor* was used (https://github.com/jlpl/nais-processor). The *nais-processor* Python package corrected for diffusional losses in the inlet (Gormley and Kennedy, 1948), applied an

ion mode calibration (Wagner et al., 2016), converted ion mobility to particle diameter, remapped the distribution
       to a new size grid, and adjusted the data to standard conditions (273.15 K, 101325 Pa).

       Further information on the removal of artefacts can be found in Sect. S2 of the supplementary material. In addition,
       a comparison between the particle concentrations measured by NAIS and the DMPS-system can be found in Sect.
       S2.5; we report a significant overestimation of particle number concentrations as measured by the NAIS in

comparison to the DMPS-system, particularly at the smallest diameters (see Fig. S9-S11). It should be noted that
       the particle number size distributions presented in this study represent that of dry aerosol particle sizes. The
       measured particle size distributions for both the NAIS and DMPS were not corrected for hydroscopic growth, and
       thus do not reflect ambient conditions.

       When assessing the qualitative and quantitative nature of NPF events, it is important to distinguish between on the

one hand the production of newly formed particles (~2-4nm) and on the other hand subsequent particle growth.
       To qualify as an event, not only does formation of ultrafine particles need to be present, but also the subsequent
       growth of the particles. Growth can last for several hours and in some cases multiple days, with the latter timescale
       suggesting that the NPF events occur on a regional scale (Kecorius et al., 2019; Ström et al., 2009). In this study,
       we distinguish between (1) *on-site formation,* defined as occasions when the concentration of 2-4nm particles

clearly rises above the background level; and (2) *formation and subsequent growth,* defined as a combination of
       *on-site formation* followed by subsequent growth. The latter is identified by visual inspection of the behaviour of
       the growing nucleation mode during the time it is present until when it disappears.

### 2.3.1 NPF Classification

       New particle formation events were classified using schemes described by Dal Maso et al. (2005) and Aliaga et

al. (2023). The classification schemes are used complementary. The classification by Dal Maso et al. (2005)
       focuses more on formation and growth, and is perhaps more intuitive to understand; however, it struggles with a
       large number of undefined cases. On the other hand, the nanoranking analysis developed by Aliaga et al. (2023)
       can be used to explore the NPF intensity and focuses on the formation of nucleation mode particles as opposed to
       the subsequent growth. The Aliaga-method classifies NPF events into 3 groups base on the overall ranking of NPF

intensity, avoiding the problem with unidentified events days (see Fig. S14 for the groups).

*Dal Maso classification system*

       In the Dal Maso et al. (2005) scheme an NPF event is defined if there exists a nucleation mode (3-25nm) which
       prevails for several hours and shows signs of growth; these events are further subdivided into three separate classes,

namely, Class Ia and Ib, and Class II. Class I and Class II events are separated on the basis of whether the growth

and formation rates, of the well-behaved size distribution, can be determined with good confidence. Class I is

further divided into Class Ia and Class Ib depending on whether there are pre-existing particles obscuring the

newly formed mode (Dal Maso et al., 2005). Undefined events are days where there are sporadic occurrences of

nucleation-mode particles, and this classification is used to separate clear events from clear non-events.

In this study, undefined days include days where the occurrence of nucleation mode particles (3-25nm) is linked

to windblown snow and situations when ZEP is enveloped by clouds. For more details regarding these sampling

conditions see Sect. S2.1 and S2.2 in the supplement for in-cloud-sampling and windblown snow events

respectively.   As a result, days classified as undefined are ones which experience bursts of nucleation mode

particles, either from windblown snow (see Fig. S4), in-cloud conditions (see Fig. S2), from non-growing NPF or

a combination of all three.   Non-events are days that display no nucleation mode (3-25nm).   It is important to note

that this scheme assumes that particle formation occurs over a geographically wide area more or less

simultaneously. Identification and classification were performed using daily surface plots of particle number size

distribution based solely on NAIS data, that is, 2.5-40nm.

It should be noted that there is a degree of subjectivity to this classification approach, as to what is considered a

certain class. The addition of the NAIS, in combination with the DMPS-system, made it easier to detect events

using surface plots, especially when the growing mode struggled to surpass ∼6nm.   However, we were not able

to identify any NPF events with the NAIS that we could not identify with the DMPS-system, as close inspection

of the number concentration of the lowest size bins of the DMPS-system was sufficient to recognise NPF events.

### *Nanoparticle Ranking Analysis*

The Nanoparticle Ranking Analysis method developed by Aliaga et al. (2023) provides a more continuous method

to characterise atmospheric NPF, allowing users to gauge the strength of NPF.   The nanoparticle ranking analysis

was utilised using the daily maximum concentration of 2.82-5nm particles ($\Delta N_{max,\,2.82-5}$), from the negative channel

of the NAIS, as the intensity parameter (note that no background value was used, unlike in Aliaga et al. (2023),

due to the low concentrations measured at ZEP). In this study, periods which experienced windblown snow or in-

cloud events were removed before applying the Nanoparticle Ranking method, as opposed to the Dal Maso et al.

(2005) approach which classified them as undefined days. Special care was taken to clean the data before utilising

the nanoparticle ranking analysis method, however it may still be the case that not all in-cloud and windblown

snow events were removed successfully (see Sect. S2.1 and S2.2 in the supplement). Windblown and in-cloud



events can influence the daily maximum without necessarily reflecting NPF activity. The strength of NPF activity was split into 3 groups depending on the intensity (i.e., *g1, g2, g3*), where *g3* is the most intense event, followed

by *g2* and then *g1;* see Aliaga et al. (2023) for more details (see Fig. S14).

### 2.3.2 Growth rates

The growth rate (GR) describes the rate of change of the modal diameter during an NPF event. For the GR estimation, NAIS and DMPS size distributions were merged at the maximum diameter of the NAIS size range (i.e. 40nm). The NAIS and DMPS data were both interpolated to 15-minute arithmetic means using linear interpolation.

The start and end of each NPF event including their sequential growth (i.e. (2) *formation and subsequent growth,*) were estimated by visual inspection. The growing nucleation mode was isolated, and the GR was estimated for the full duration of the NPF event (see Fig. S15). For each size bin, we applied a Gaussian filter (sigma = 1) and found the time in which the concentration of a particular size bin experienced the highest maximum rate of change. Essentially, the evolution of the nucleation mode was tracked using the timings for each identified peak in the rate

of change of the growing nucleation mode diameter. The GR was calculated using a simple linear ordinary least squares (OLS) method.

It should be noted that the aforementioned issues concerning the overestimation of particle number concentrations for the NAIS measurements do not impact the calculation of GR as it only examines the changes to the diameter of the nucleation mode. Very few events grew beyond the limit of the NAIS (i.e. 40nm), hence the impact of a

diameter jump from NAIS to DMPS was not considered. We tried to measure the GR of all NPF events (e.g. Class 1a, 1b and II), even though by definition Class II events lack the strength and consistency, making it difficult to calculate GRs (Dal Maso et al., 2005). The GRs for some Class II events were not estimated due to uncertain tracking of the nucleation growth.

### 2.3.3 Particle Formation Rates

The formation rates namely $J_{3-7}$, $J_{7-25}$, and $J_{3-25}$ ($cm^{-3}s^{-1}$) were estimated using the following equation:

$$J_{i-j} = \frac{dN_{i-j}}{dt} + \text{CoagS} \times N_{i-j} + \frac{\text{GR}}{\Delta d_p} \times N_{i-j}, \qquad (1)$$

where $J_{i-j}$ is the formation rate of particles between diameters i-j nm, CoagS is the coagulation sink of the pre-existing particle population, GR is the growth rate between i-j nm and Δdp is the difference in diameter between i and jnm.



The Python GUI, *npf-event-analyzer*, was used to provide additional complementary estimations of the GRs and

formation rates for the NPF events (i.e. in addition to the OLS fitting method) (**https://github.com/jlpl/npf-event-analyzer).** The max concentration method was used for the estimations of GR and J.

### 2.3.4. Condensation sink

The condensation sink (CS) is defined as the rate at which non-volatile vapours condense onto pre-exiting particles

(Kulmala et al., 2012). The twin DMPS system (5-850nm) was used to calculate CS, for SA vapour. The method

for the determination of CS is described by Dal Maso et al. (2002), and uses the following equation:

$$CS = 2\pi D \int_{i}^{d_{p,i\,max}} d_{p,i}\beta(d_{p,i})N(d_{p,i})\,\mathrm{d}d_{p,i},$$      *(2)*

where $d_{p,i}$ is the diameter of a particle in size class $i$, $N_i$ is the particle number concentration in the respective size

class, and $D$ is the diffusion coefficient of the condensing vapour (in this case condensing vapours were assumed

to have $H_2SO_4$ diffusion properties). We used the transition regime correction factor $\beta$ from Fuchs and Sutugin

(1970).

**2.3.5 Concentration of condensable vapours and source strength**

The concentration of vapours (Cv) required to sustain the calculated GRs and their respective source rates (Q)

were estimated using the following equations, and assuming the condensation of $H_2SO_4$.

$$C_v = A \times dD_p/dt.$$      *(3)*

where A is the constant $1.37 \cdot 10^7$ hcm$^{-3}$ nm$^{-1}$ representative for the molecular properties of SA, $dD_p/dt$ is the rate

of change of the nucleation mode i.e. GR, and CS is the condensation sink. See Dal Maso et al. (2005) for more

detail.

### 2.4 Airmass analysis

### 2.4.1 Transport model

Airmass back trajectory analysis was performed using the Hybrid Single-Particle Lagrangian Integrated Trajectory

model (HYSPLIT V5.2.1) (Draxler et al., 1998; Stein et al., 2015). Ensemble back trajectories were initialised

every hour starting at the latitude and longitude of ZEP, and at a height of 250m. The ensemble was generated by

offsetting the meteorological grid point by one in the horizontal (i.e. dxf = 1.0, dyf = 1.0) and by the default offset

of 0.01 sigma units in the vertical (~250m), thus generating 27 back trajectories for all possible offsets in X, Y

and Z. The starting height of 250m ensured that the starting location of all the ensemble members was at the surface or above. The back trajectories were calculated for 5 days back in time. Global Data Assimilation System

(GDAS) 1°×1° archive data (http: //ready.arl.noaa.gov/archives.php, last access: 08 08 2024) was used as the meteorological fields.

The accumulated solar flux in the most recent 6 hours prior to arrival was used for the solar insolation term used throughout this study. The solar flux was taken from the original GDAS-derived HYSPLIT output. Furthermore, only endpoints within the mixed-layer were used as this is when air masses are influenced by the surface. The last

6 hours were used, as opposed to a longer duration, to place more significance on solar radiation closer to the receptor; an accumulated solar flux consisting for a longer duration could lead to misleading results.

### 2.4.2 Chlorophyll a data

Daily mapped chlorophyll a data derived from satellite observations (Aqua MODIS) (last accessed: 2024-10-04, downloaded from https://oceancolor.gsfc.nasa.gov/l3/) were utilised along with the HYSPLIT output to estimate

the chlorophyll a exposure. Exposure of the air masses to chlorophyll a was calculated based on the equation developed by Park et al. (2018), which linked chlorophyll a with DMS measurements on site at ZEP. Chlorophyll a exposure was essentially used as a proxy for DMS emissions. DMS in turn can form $H_2SO_4$ via $SO_2$, although this was not explicitly addressed by Park et al. (2018). Park et al. (2018) utilised the relationship between DMS and air mass exposure to chlorophyll a to account for the differences in the species-specific phytoplankton

population, between the Greenland Sea and the Barents Sea.

### 2.4.3 Nucleation site estimation

The method utilised here is similar to the NanoMap analysis developed by Kristensson et al. (2014) in which the geographic position of where new particle formation took place was estimated based on the particle number size distribution measurements during NPF events and HYSPLIT back trajectories. In this study, the duration of the

nucleation mode was calculated. For every hour of the NPF event, back trajectories were initialised. The length at which the back trajectories were initialised for depended on how long the event had lasted. In short, $i$ hours after the event begun, the back trajectory will have been initialised to run $i$ hours backwards. So, for every hour except the start time the back trajectories were initialised for the following length of time: *back trajectory length = duration - (end time - t),* where $t \in$ (end to start times). Hence, for an NPF event that lasts 7 hours, 6 back trajectory

ensembles were calculated for varying lengths from 7 hours to 1 hour long.

### 2.4.4 Air mass marine regions

The endpoints for every 5-day back trajectory ensemble were assigned to a marine region e.g. Arctic Ocean, Greenland Sea, Barents Sea, Norwegian Sea and North Atlantic Ocean (see Fig. S18 in supplement for defined regions). Days were assigned to a specific marine region based on the cumulative residence time in the specific

sector. A threshold of 0.5 (50%) was used as qualifier. Days without clear source preference, i.e. the airmasses that did not reach the 50% threshold for either of the pre-defined source regions, were classified as mixed.

## 3. Results & Discussion

### 3.1 Seasonality and interannual variability

#### 3.1.1 Annual cycle of NPF events

The 848 days of valid NAIS measurements (2022-04-17 to 2024-10-04) are classified according to the Dal Maso et al. (2005) classification system. 157 days are classified as NPF events, either Class Ia events (28), Class Ib (19) or Class II (110). A total of 438 days are classified as Undefined, and the remaining 271 days are considered to be either Non-event days (253) or consisting of too little data (i.e. defined as <12 hours of data per day) (18) (see Fig. 1). Overall, this meant that 19% of days with valid NAIS measurements experience an NPF event (i.e. Class Ia,

Ib or II). The majority of days (52%) are classified as undefined due to the appearance of a non-growing nucleation mode that persisted for some hours. The undefined days are further sub-divided into those containing windblown snow events (19% of all data), in-cloud events (8% of all data) (see Fig. S6 in the supplement). Of the 438 Undefined days, 72 are a result of non-growing bursts (i.e. 17% of the Undefined days).

The monthly frequency of NPF events (i.e. class Ia, Ib or II) shows events first appearing in April (see Fig. 1),

reaching a peak, in terms of the occurrences per month, in May to July with approximately 30 - 40% of days experiencing NPF. The daily occurrence decreases during the month of June compared with the preceding and succeeding months, a consistent finding throughout the three years of data. After July, the frequency of events declines towards the end of summer and into autumn, where NPF events cease to be present beyond November. NPF events were observed as late in the year as November, which is interesting given that this is during the polar

night when the Sun stays below the horizon (as witnessed by ZEP) and thus ZEP experiences negligible solar radiation. Multiple NPF events were observed during the polar night period (see Sect. 3.1.3 for more details). April 2022 appeared anomalous perhaps given that the measurements for that month were not an entire month as the measurements started on 17th April 2022.



Prior studies, at ZEP, which used slightly different event classification schemes, reported the occurrence of NPF

events to be within the range of 18%-23% (Dallósto et al., 2017; Heintzenberg and Leck, 2012; Lee et al., 2020).

Nieminen et al. (2018) reported NPF day frequencies of 14% (Mar-May), 34% (Jun-Aug) and 6.6% (Sep-Nov)

and 0% (Dec-Feb). Lee et al. (2020) observed the highest frequency in terms of the formation and growth of $N_{3-25}$ particles in June (>40%), with the NPF frequency in May and July being ~40% and <40%.

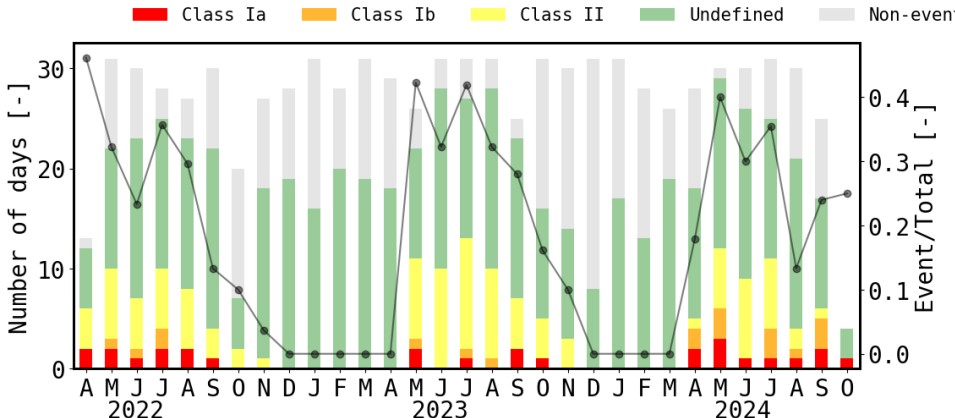

**Fig 1: Classification of almost three years of Neutral cluster and Air Ion Spectrometer (NAIS) particle number size distribution using the Dal Maso et al. (2005) classification system. Days in which there were no measurements were not included. Event/Total was the ratio of event days (i.e. Class Ia, Ib, and II) over the total number of days (i.e. Class Ia, Ib, II, Undefined and Non-event).**


### 3.1.2 Interannual variability of main parameters

The interannual variability of the following main parameters, namely NPF intensity, solar insolation, DMS concentration and the CS, are explored in figures 2a-b. A proxy for the intensity of events, as opposed to the frequency, was determined by utilising elements of the nanoparticle ranking analysis developed by Aliaga et al.

(2023). The maximum daily increase in $2.82 - 5nm$ particles ($\Delta N_{max\ 2.82-5}$), was used to represent the strength of NPF activity (shown to correlate well with calculated J). See Sect. 2.4.1 in the Methods for more details concerning the Nano ranking method and about the definition of solar insolation.



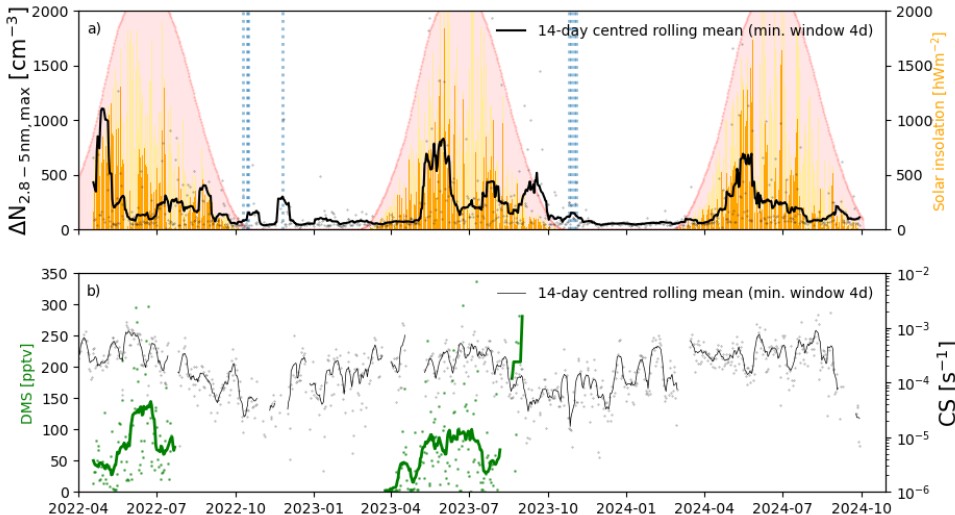

**Fig 2:** Timeseries a) presenting in black dots the intensity parameter, i.e. the daily maximum total number concentration of particles between 2.82 and 5 nm ($\Delta N_{max, 2.8-5}$), which corresponds to the strength of the atmospheric NPF event. The black line signifies the running 14-day (centred) mean for $\Delta N_{max, 2.8-5}$ and was applied with a minimum window of 4 days. The orange bar chart displays the daily-averaged solar insolation, which was defined as the accumulation of the most recent 6 hours of solar radiation experienced by air masses arriving at ZEP, within the mixed-layer (see Sect. 2.4.1 in the Methods for details). The faint yellow bars are for 5 days, regardless of if airmasses were within the mixed-layer. The shaded region in red is the average hourly clear skies global horizontal irradiance (GHI) (calculated based on the altitude, latitude and longitude of ZEP, using Python Package *pvlib* v0.9.0 https://pvlib-python.readthedocs.io/en/v0.9.0/api.html#clear-sky) multiplied by six. The timeseries b) displays all valid dimethyl sulphide (DMS) measurements recorded at ZEP in green (typically, the instrument begins operation around late March or early April). In black, the condensation sink (CS) at ZEP is displayed, including a running mean (line) and daily means (dots). For the CS and DMS, a 14-day running (centred) mean is used with a minimum window of 4-days. The dashed blue lines represent the occurrence of the polar night events.

$\Delta N_{max\ 2.82-5}$ shows various peaks throughout the annual cycle, however, the one striking feature is that $\Delta N_{max\ 2.82-5}$

the first and highest peaks in $\Delta N_{max\ 2.82-5}$ occur in late spring and the beginning of summer (i.e. end of April in 2022, beginning of June in 2023, and towards the end of May in 2024) (see Fig. 2a)). For the annual cycle in terms of frequency (see Fig. 1), the first peak in occurrence matches the increased $\Delta N_{max\ 2.82-5}$ (Fig. 2a)). However, the succeeding peak in frequency (i.e. in July) coincides with reduced intensity ($\Delta N_{max\ 2.82-5}$). The seasonality of the $\Delta N_{max\ 2.82-5}$ value follows that of the solar insolation, and for the years 2023 and 2024, the maximum NPF strength

(i.e. $\Delta N_{max\,2.82-5}$) coincides with the greatest mean solar insolation; a much better relationship is observed when selecting for air masses within the mixed layer as this is when arriving air masses are influenced by the surface. The annual cycle for the calculated CS exhibits a sustained maximum in late winter and early spring due to the Arctic Haze, and the CS stays high into the summer before reaching a minimum in autumn, typically the cleanest part of the year (Tunved et al., 2013) (see Fig. 2b)). During the Arctic Haze period, long-range transported air

masses from Eurasia bring aged aerosol contributing to higher CS. At the same time this period is characterised by higher sulphate and $SO_2$ concentrations, mainly of anthropogenic origin (Platt et al., 2022). One additional aspect is that it is clear that increases in $\Delta N_{max\,2.82-5}$ occur during periods where there was a slight decrease in CS, below that of the seasonal CS average. For example, the peak in $\Delta N_{max\,2.82-5}$ in April in 2022 (see Fig. 2b)), which occurs slightly prior to the maximum solar insolation, can potentially be explained by the reduction in the CS

during that period.

The solar intensity is a controlling factor in the production of SA (i.e. $H_2SO_4$) via the oxidation of $SO_2$ (among other potential precursor candidates), whilst the CS acts as a sink for the newly-formed particles, and the coagulation sink (CoagS, CS can act as a proxy for CoagS) acts a sink for nucleating vapours, and thus inhibit NPF. NPF intensity ($\Delta N_{max\,2.82-5}$) is skewed, with a greater intensity towards the end of spring and beginning of

summer. Similarly, solar insolation is also shifted towards the start of summer, even though the theoretical solar maximum has a symmetrical distribution centred around the middle of summer. The skewed pattern in both the solar insolation and NPF intensity ($\Delta N_{max\,2.82-5}$) can perhaps be explained by changes in cloud cover (see Maturilli and Ebell (2018) for frequency occurrence of cloudy sky). Late autumn experiences numerous, but relatively weaker, NPF events, with peaks in $\Delta N_{max\,2.82-5}$ (~100-500cm$^{-3}$) every year at the end of the summer, despite the

relatively small solar insolation. The transition period between summer and haze (i.e. Oct. – Jan.) is typically the cleanest part of the year, and thus exhibits the lowest CS (Tunved et al., 2013). During this part of the year the small amount of solar insolation and vapours present are enough to trigger NPF, considering the low CS.

Finally, DMS is the main of source of sulphur for clean marine aerosol particles, and constitutes an important compound in regards to NPF via its oxidation products. Numerous peaks in DMS are observed during the

summertime, with a maximum observed in July (see Fig. 2b)). Elevated concentrations of DMS at ZEP coincide with air masses which traverse over marine regions containing chlorophyll a (not shown here, but also noted by Park et al. (2018b)).



### 3.1.3 Events with minimal solar insolation (i.e. Polar night events)

Contrary to expectations, during the polar night (i.e. when the sun does not cross the horizon at Ny-Ålesund) a

number of NPF events are observed. As shown earlier, solar insolation is clearly one of the driving parameters for
NPF occurrence and intensity, and Ny-Ålesund in November does not receive any direct solar radiation (see Fig.
S16 for examples of the polar night events). For the November polar night events, air mass back trajectories
indicate that air masses have to travel at least half a day backwards before reaching Ny-Ålesund to experience
direct solar radiation.  In total, five types of these events are observed, i.e. 1. 2022-11-25 - 2022-11-26, 2. 2023-

10-25 - 2023-10-26, 3ab. 2023-10-28 (x2, 01:00 12:00) – 2023-10-29, 4. 2023-11-01 - 2023-11-02, 5. 2023-11-02
-2023-11-03.  The polar night NPF event starting on the 2023-10-25 appears to be a transported event (TE, see
Dada et al. (2018) for details on the definition of a transported event) as we do not observe the entire growth of
freshly nucleated particles; instead the growing nucleation mode begins at 4 nm.    We find that these events
coincide with air masses mainly arriving from lower latitudes (i.e. further south). The extremely low accumulated

solar radiation experienced by the arriving air masses during these events means that there should be little available
nucleating vapours (less oxidation before reaching ZEP).  Two other Arctic sites namely Utqigvik, Alaska and
Tiksi, Russia, also observe NPF events without the sites experiencing any solar radiation (Asmi et al., 2016;
Kolesar et al., 2017b).  However, in these two cases the suggestion is that anthropogenic emissions of semi-volatile
gases from nearby oil fields or the build-up of anthropogenic emissions during the haze period can explain these

events; an interpretation which is not applicable for ZEP.

The vertical component of the arriving back trajectories associated with these polar night events shows that air
masses originated from higher altitudes (Figure S17, in the supplement), and that theses air masses descend before
arriving at ZEP.  One possible explanation for these types of events could be that precursor gases, originally
emitted from the surface, are transported to high altitudes where the CS is sufficiently low enough to allow them

to survive long enough to be nucleated on decent to ZEP.  The air masses travel further south and higher in altitude,
where the small amount of solar radiation potentially allows for photochemical production of nucleating vapours
to build up.

### 3.1.4 Start times and duration throughout the year

The median duration of NPF events, including their subsequent growth, is approximately ∼10 hours and seems to

be fairly consistent across the annual cycle, with a decrease towards the autumn (see Fig. 3a)). Of all events, 75%
> 7 hours, and 90%> 4 hours.  During the sunlit period, the starting time of events is typically around 09:00 (UTC)
for the period April to August. From September onwards, the onset of the observed NPF shifts to later in the day



(Fig. 3b)). For the period August to October, onset of NPF is approximately 8 hours after sunrise at Ny Ålesund (see the gradient of the two slopes in Fig. 3b) remains constant but the start time is shifted by 8 hours). This reflects

perhaps that during the NPF season there is a delay of approximately 8 hours from solar intensity minimum to the onset of NPF, and might be related to the time it takes for the availability of precursor gases to build up in the mixed layer.

Lee et al. (2020) suggested that the duration of NPF at ZEP was approximately 6-7 hours, with the longest duration in summer. In addition, NPF start times were 13:00 – 14:00 (local time, which is 11:00 – 12:00 UTC in the summer

time). The start times observed at ZEP, here in this study, were typically 2 hours earlier, perhaps as a result of the slightly lower minimum particle diameter detection and not the same definition of the onset of nucleation.

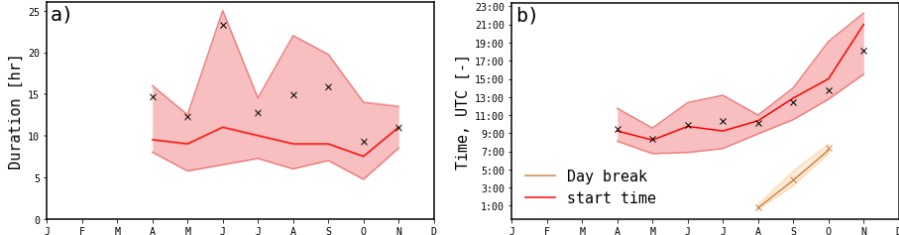

**Fig 3: (a) the duration of the entire growth of newly formed particles from the first measurable size bin i.e. 2.5nm to the largest size bin (b) the average start time of the observed new particle formation (NPF) (monthly mean displayed by the cross and median by the red line), along with a shaded region representing the 25th and 75th percentiles. Along with the estimated time the global horizontal irradiance (GHI) goes beyond zero (i.e. daybreak). The average time of day break for the months April until July is not applicable, since the sun is constantly above the horizon (as experienced by ZEP). Time is given in terms of coordinated universal time (UTC).**

### 3.1.5 Seasonality of Growth Rates and formation rates

The average, mean (median), GRs for all the events is 2.2 (1.5) nmhr$^{-1}$. For class Ia, Ib, and II events the average GRs are 1.4 (1.0), 1.7 (1.2), 2.6 (1.2) nmhr$^{-1}$ respectively. The largest GRs occur during July 4.0 (2.1) nm hr$^{-1}$. The

polar night events exhibit a significant reduced GR of 0.43 (0.35) nmhr$^{-1}$ (see Fig. 4).    Using the *npf-event-analyzer* for the Class Ia events the GRs are 0.9 (1.1), 2.1 (2.6), 4.1 (4.4) nmhr$^{-1}$ for 3-7, 3-25 and 7-25 nm diameter ranges.

The GRs presented here agree well with earlier estimates of GRs by Nieminen et al. (2018) reporting a GR$_{10-25nm}$ at ZEP within the range of 1.2 -1.6 nmhr$^{-1}$.  Kerminen et al. (2018) report a median GR for the Arctic of 2.3 (0.23-

4.1, 5th and 95th percentiles) nm hr$^{-1}$, the site type (i.e. globally) with the smallest GRs.





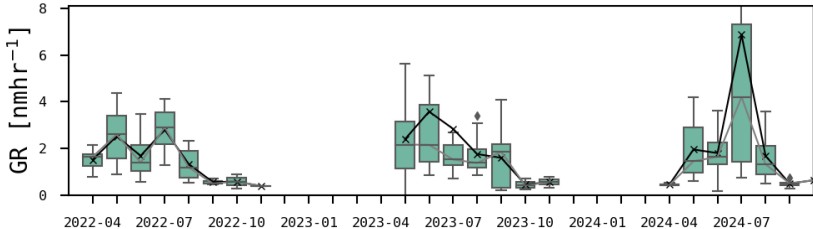

**Fig 4: The calculated growth rates (GRs) throughout the measurement period are represented by box-whiskers, which display the monthly growth rate averages and include the median and 25[th] - 75[th] percentiles. The green box-whiskers refer to the average GRs from the start of the NPF until it stops growing. The black line represents the mean.**

The formation rates for $J_{3-7nm}$, $J_{3-25nm}$ $J_{7-25nm}$ are estimated to be 0.28 ± 0.33, 0.40 ± 0.35 and 0.31 ± 0.28 cm$^{-3}$s$^{-1}$ (mean ±std) for class Ia events (using the max concentration method and *npf-event-analyzer* see Sect. 2.3.3 in the Methods).

Previous studies also estimate J for various diameters at Ny-Ålesund and ZEP. Beck et al. (2021) reports a $J_{1.5}$ equal to 0.27 cm$^{-3}$s$^{-1}$ (Beck et al., 2021). Nieminen et al. (2018) presents the seasonal medians for $J_{nuc}$ of nucleation-mode particles (10-25nm) varying from 0.08 cm$^{-3}$s$^{-1}$ in spring, to 0.032 cm$^{-3}$s$^{-1}$ in summer and to 0.0066 cm$^{-3}$s$^{-1}$ in autumn. Lee et al. (2020) gives $J_{3-7}$, $J_{7-25}$, $J_{3-25}$ values ranging between 0.001-0.54, 0.003-0.5, 0.007-0.61 cm$^{-3}$s$^{-1}$ respectively, and the averages for $J_{3-7}$, $J_{7-25}$, $J_{3-25}$ are 0.04, 0.09 and 0.12 cm$^{-3}$s$^{-1}$. Kerminen et al. (2018)

report a median Arctic J (i.e. varying size ranges) of 0.51 cm$^{-3}$s$^{-1}$.

In comparison to GRs and Js at other sites (e.g. Kerminen et al. (2018); Nieminen et al., (2018)), we see that processes within the Arctic, as observed at ZEP, are significantly slower. In urban environments, compared with more remote ones, estimations for J are significantly larger due to the larger coagulation sink that needs to be overcome (Cai and Jiang, 2017); here, the coagulation sink is much lower than in urban environments.

**3.1.6 Seasonality of Condensable Vapours**

The concentration of condensable vapours (Cv) required to sustain the GRs is estimated using equation (3) and the calculated growth rates for the various classes of events (i.e., Class Ia, Ib and II). We show that Cv increases until July before decreasing during the autumn and into the winter months. The seasonal variation is similar to that of the sum of the measured concentrations of potential aerosol precursor gases found in the high Arctic during the

MOSiAC campaign (Boyer et al., 2024), namely sulphuric acid (H$_2$SO$_4$), methanesulphonic acid (MSA) and iodic

acid (HIO$_3$). The sum of these three species increases during the spring, peaks in May and then decreases towards the end of the year (see Fig. 5). Overall, the concentration of the calculated Cv is on average 5.9 times greater than the combined concentrations of the measured precursor gases (H$_2$SO$_4$, MSA and HIO$_3$). For Ny-Ålesund, Beck et al. (2021) report increasing IA, SA and MSA concentrations from the start of the sunlight period, and high

concentrations prevailing before decreasing around mid-June. HOM concentrations are very low during spring and increase in May, however it still shows their importance in contributing to Cv.

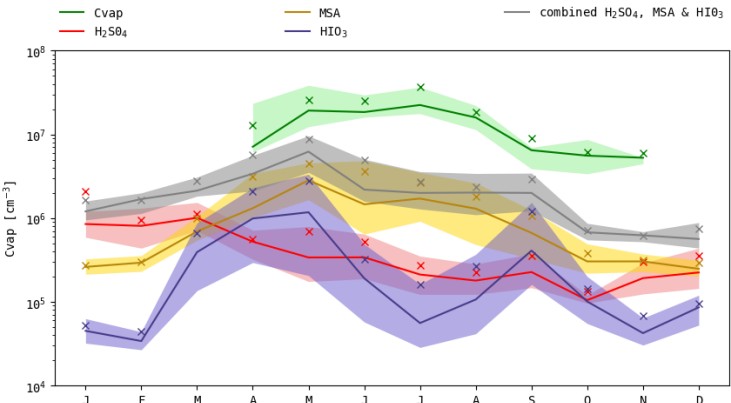

**Fig 5: The monthly means (crosses) and medians (thick line), along with the 25[th] and 75[th] percentiles (shaded region) for the estimated concentration of equivalent condensational vapours (Cv, green), and the concentrations of measured high Arctic precursor vapours namely sulphuric acid (H$_2$SO$_4$, red), methanesulphonic acid (MSA, gold) and iodic acid (HIO$_3$, purple) and combined (grey) from (Boyer et al., 2024), covering the years 2022 to 2024.**

Considering the concentrations of precursor vapours from Boyer et al. (2024) are measured at higher latitudes (>80°N), where there is less photochemistry and biological activity, the calculated Cv and combined measured

concentrations of precursors are not too different in magnitude. The sources of available precursors are considered to be more limited where the MOSiAC campaign was sampling (Brean et al., 2023), and may be more influenced by the sea ice cover. Furthermore, the calculated Cv presented here is only available for days with NPF, while the results from Boyer et al. (2024) represent average concentrations regardless of if NPF is observed or not. Hence, it is natural to expect higher concentrations than when NPF takes place, which in turn may explain as to why the

calculated Cv is higher than the presented observations. Also, it cannot be excluded that other species other than those presented in Boyer et al. (2024) partake in the growth of the nucleation mode.





### 3.2 Positive and negative ions during NPF

The polarity of air ions, which aid in nucleation, provides information about the composition of the newly formed aerosol particles. Cluster air ions are typically defined between 0.8-1.7nm, whilst intermediate ions range from

1.6-7.4nm. Intermediate ions are created in the initial stage of nucleation and are typically formed when neutral nanoparticles encounter cluster ions and acquire their charge (Tammet et al., 2014). Negative ions can be linked to IA and SA to name a couple of compounds.

The total number of positive ($N^+_{2-4nm}$) and negative ($N^-_{2-4nm}$) small intermediate ions (2-4nm) within the growing mode are calculated for each NPF event. In May, at the start of the NPF season, we observe the highest ratio of

negative small intermediate ions compared to small positive intermediate ions (i.e. mean and median $N^-_{2-4nm}/N^+_{2-4nm}$ = 4 and 1.4 respectively). Higher $N^-_{2-4nm}/N^+_{2-4nm}$ ratios perhaps relate to the presence of negatively charged molecules participating in the initial stages of ion-induced nucleation. Moreover, the ratio is higher during months that experience greater NPF intensity (i.e. larger production of nucleation mode particles). Towards the end of the NPF season (i.e. Sep. - Nov.) there is a shift towards a growing nucleation mode (2-4nm) more dominated by

positive small intermediate ions, as opposed to negative small intermediate ions (see Fig. 6). The tendency for small intermediate ions (2-4nm) to exhibit a positive charge appears from September onwards, which is also when there begins to be limited solar insolation and the occurrence of polar night events (i.e. late Oct. and Nov.). The consistent loss of the cluster ions in the negative polarity (see Sect. S2.6 in the supplement) means that the ratio between the concentrations of the two polarities for ions within that size range (i.e. <2nm) is not explored. Despite

this, the shift in the overall charge of the small intermediate ions, from more negative to more positive, potentially hints at a seasonal change in the vapours contributing to the initial formation of critical clusters.

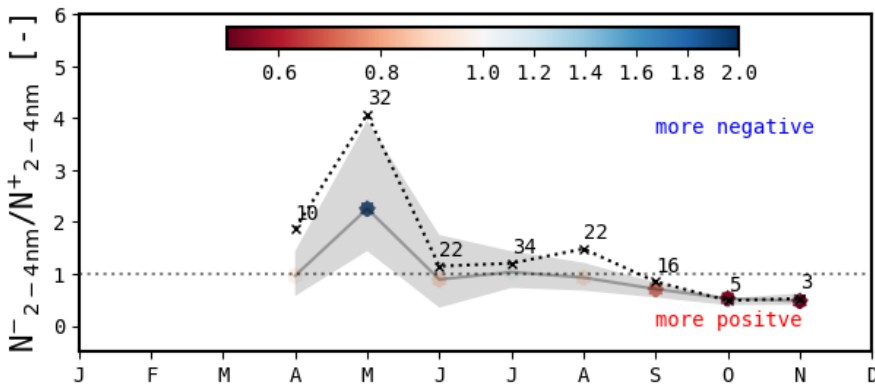



**Fig 6: For all new particle formation (NPF) events the monthly ratio of the total number concentration of negatively charged small intermediate ions over positively charged small intermediate ions (2-4nm) ($N^-_{2-4nm}/N^+_{2-4nm}$). $N^-_{2-4nm}/N^+_{2-4nm}$ includes only ion concentrations within the identified growing mode between 2 and 4nm. The mean monthly $N^-_{2-4nm}/N^+_{2-4nm}$ is presented as a cross, and the median is displayed by coloured circles, blue for an overall negative charge (ratio>1) and red for an overall positive charge (ratio<1). The grey shading displays the $25^{th} – 75^{th}$ percentiles. The number of events which make up the monthly values are written above each respective cross.**

As mentioned above, Beck et al. (2021) reports higher concentrations of precursor gases with low proton affinity (e.g., MSA, SA, $HSO_4^-$) during the month of May, while highly oxygenated organic molecules (HOMs) display higher concentrations than SA and MSA during April and from June towards September. Beck et al. (2021) suggest that the events in May were likely initiated by negative ion-induced NPF and relate to sulphuric acid and ammonia nucleation. The ratio seems to reflect the varying seasonal NPF strength (i.e. production of nucleation mode particles). It should be noted that there may be some instrumental biases between the two polarities, however, the background concentrations (during no-event periods) display $N^-_{2-4nm}/N^+_{2-4nm}$ ratios close to unity, ~1.02. Ammonia ($NH_4^+$) is a possible positively charged cluster ion which could play a role in the shift in polarity (Kirkby et al., 2016), however, it needs to be stressed that further work is needed to ascertain it there is a shift in the polarity of the air ions.

### 3.3 Source regions and geo-spatial extent

NPF events are assigned to various marine regions using the method described in Sect. 2.4.5. Air masses that traverse over the Arctic Ocean coincide with the highest number of NPF events (57 in total or 37% of NPF events; see Table 1). 37% of air masses are classified as originating from the Arctic Ocean, the largest marine region in terms of contribution. The air masses that mainly originate over the Greenland Sea result in 29 NPF events and show the highest probability of an NPF event occurring, with 27% of the Greenland Sea air masses being linked to an NPF day. Air masses originating over the Arctic Ocean and Barents Sea have a 19% and 13% likelihood, respectively.

*Table 1: The frequency of occurrence of air masses from different sectors compared with the number of events. Both numbers are given in days, followed by a ratio of the two (i.e. Likelihood). Although not shown in table, back trajectories traversed over the Kara Sea on two occasions. The average CS, solar insolation (both mixed-layer and all), maximum daily*





*increase in 2.82 − 5nm particles (ΔN<sub>max, 2.82-5</sub>) and proportion of back trajectory within mixed layer are given for each*
*marine sector. The averages for these variables area also given during the NPF days and presented in brackets.*

| Air mass | Region's frequency (days) | Number of NPF events (days) | Likelihood | CS [s-1] | Solar insolation [Whm$^{-2}$] | ΔN$_{max, 2.82-5}$ [cm$^{-3}$] | Proportion in ML |
|---|---|---|---|---|---|---|---|
| **Arctic Ocean** | 313 | 57 | 0.19 | 0.20 (0.19) | 185 (378) 368 (753) | 239 (776) | 0.63 (0.60) |
| **Greenland Sea** | 109 | 29 | 0.27 | 0.25 (0.25) | 327 (442) 666 (991) | 254 (665) | 0.60 (0.54) |
| **Barents Sea** | 69 | 9 | 0.13 | 0.36 (0.29) | 118 (211) 597 (1100) | 93 (204) | 0.32 (0.28) |
| **Norwegian Sea** | 44 | 7 | 0.16 | 0.25 (0.43) | 232 (418) 618 (1330) | 136 (382) | 0.48 (0.34) |
| **Mixed** | 303 | 51 | 0.17 | 0.23 | 184 | 147 | 0.54 (0.50) |
| **Total** | 840 | 153 | 0.18 | 0.23 | 200 | 190 | 0.57 (0.53) |

The analysis is expanded, using the method outlined in Sect. 2.4.4, by estimating the location of the formation of newly formed particles. The most common location for nucleation to occur is off the western coast of Svalbard, i.e., in the Greenland Sea and Arctic Ocean (see Fig. 7). Furthermore, the Greenland Sea, south-west of ZEP, as

opposed to the Barents Sea, is home to more nucleation sites. It is also clear that the vast majority of nucleation sites are estimated to occur over the ocean. The estimated nucleation sites are almost all within a $1 \cdot 10^6$ km radius of ZEP, and the vast majority are fairly close to ZEP (see Fig. 7), a finding which can also be inferred from the observation that the majority of events begin growing from 2.5nm. Furthermore, from the air mass back trajectory analysis, we gather that air masses typically reside between sea level and the altitude of ZEP (i.e. 0-474m, see Fig.

S17). There is little difference between the altitudes of the arriving air masses for the NPF events compared to all days.



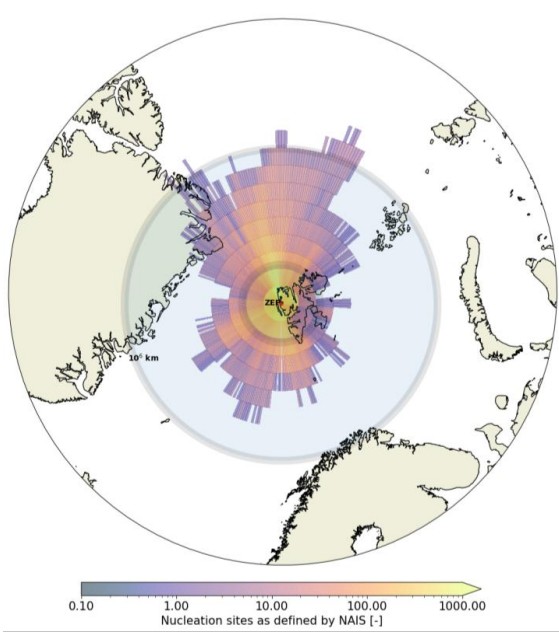

**Fig 7: Total number of estimated nucleation sites per grid cell. Unit is based on the number of back trajectories initialised and does not reflect a physical value. Sites of nucleation for class Ia, Ib and II events are estimated by using HYSPLIT to trace the likely origin of particles produced during NPF events, and initialising the back trajectories based on the duration of the NPF event (see Sect. 2.4.4).**

DMS emissions are closely linked to phytoplankton dynamics and abundance, and typically occur following phytoplankton biomass maxima (Galí and Simó, 2015). Numerous studies have tried to link chlorophyll a, which

is used as a proxy for phytoplankton biomass, with DMS emissions to the atmosphere (Jang et al., 2019). At ZEP, Park et al. (2018) note a strong correlation between DMS concentrations measured at ZEP and air masses traversing over the surrounding ocean with high chlorophyll a. They find that the relationship between DMS concentrations and air masses likely exposed to high DMS emissions are stronger for air masses traversing over the Greenland Sea, compared with the Barents Sea. Therefore, suggesting that the Greenland Sea has a DMS

production capacity three times greater than the Barents Sea.  By replicating the method detailed in Park et al. (2018), we also find good correlations between chlorophyll a exposure and DMS, however, not for all years, and not for the years related to this study (i.e. 2022 and 2023). In the study by Park et al. (2018) the years 2010, 2014 and 2015 were included, however, 2022 and 2023 did not yield such strong correlations.  We argue that because of the large spatial and temporal interannual variability, possible geo-spatial changes in the DMSP-rich

phytoplankton species in the neighbouring seas, a wide range of DMS oxidation timescales depending on photochemistry and available solar insolation (Ghahreman et al., 2019) and the influence of various other meteorological and environment factors, we are unable to replicate the findings presented in Park et al. (2018) for the years 2022 and 2023 (not shown here). In addition, direct correlations between atmospheric DMS and chlorophyll a exposure are not detected around Iceland, however, a high atmospheric DMS to chlorophyll a

exposure coincides with areas containing DMSP-rich phytoplankton species (Lee et al., 2023). By averaging over large enough areas and time periods, distinctive DMS-to-chlorophyll exposure ratios may present themselves for different marine source regions, as the time scales involved in the oxidation of gaseous precursors influence the results less.

The Greenland Sea is connected to a higher likelihood of NPF events; compared with the Barents Sea there is a

distinct difference between both seas in terms of the proportion of air masses coinciding with NPF events, and the number of nucleation sites estimated to arise from each sea (see Fig. 7). The combined effect of a larger number of air masses coming from the west and an increased NPF-likelihood linked to western air masses means that we observe more NPF events off the western edge of Svalbard. This difference could be linked to the finding that the DMS production capacity in the Greenland Sea is estimated to be three times greater than that of the Barents Sea,

due to more DMSP-containing phytoplankton species (e.g. Phaeocystis) (Park et al., 2018). However, another distinct difference between these two source regions is that air masses, which traversed over the Greenland Sea, experienced more solar insolation and a relatively lower CS average, compared to the air masses coming over the Barents Sea (see Table 1). The combination of both high solar insolation and relatively low CS, could also explain why NPF likelihood is higher for the Greenland Sea as opposed to Barents Sea. One further difference is that the

Barents Sea has a much larger sea ice coverage compared with the Greenland Sea, and thus there is potentially less open ocean in contact with the lower atmosphere during the NPF season. The reduced availability of DMS given the greater sea ice coverage could additionally help explain the differences in NPF likelihood of the Greenland and Barents Seas.

**4.1 Predicting NPF likelihood with condensation sink and solar radiation**

Above, we show that NPF events and increases in NPF intensity, typically occur during periods of high solar insolation. High solar insolation seems to be a prerequisite for NPF, as it leads to the photochemical production of nucleating compounds. NPF events also seem to require the total surface area of pre-existing aerosol particles (i.e., CS) to decrease below seasonal averages (see Fig. 2). The NPF events require nucleating vapours to reach higher than critical concentrations, and as the CS acts as a sink of those highly condensable vapours, low CS values



promote the potential for nucleation. In this section we argue that solar insolation and CS can be used as the two

central parameters determining with high certainty the likelihood that a given day will experience an NPF event.

For the definition of the term solar insolation refer to Sect. 2.4.1 in the methods.

The majority of measurements feature the combination of both relatively low solar insolation and low CS

($<1000 hWm^{-2}$ and $<0.4\cdot10^{-4}$ see bottom-left of Fig. 8a)), and on very few occasions there are measurements with

a combination of high solar insolation ($>1000 hWm^{-2}$) and low CS ($<2\cdot10^{-4}$) (see top-left of Fig. 8a)).

Nonetheless, we show that when the combination of high solar insolation ($>1000 hWm^{-2}$) and relatively low CS

($<0.4\cdot10^{-4}$) (i.e. top-left of Fig. 8b)) arises, it occurs during an NPF day (as defined by Aliaga et al. (2023)). We

find that periods exhibiting high solar insolation ($>1000 Wm^{-2}$) and relatively low CS ($<4\cdot10^{-4}$) are linked to more

NPF days. Overall, the combinations of CS and solar insolation which promote NPF are relatively rare, compared

to the entire set of measurements. The probability of occurrence is derived from the ratio of the number of

measurements for each combination of CS and solar insolation during an NPF day to the number of valid data

points (hourly) for each of these combinations (see Fig. 8b)).

We define a region of interest (ROI) (represented using the dashed line, separating the top left-hand corner in

figure 9b). The ROI represents a region where the likelihood that a set of measurements is linked to an NPF day

is on average, mean (median), 66.3 (68.4) % (ROI consists of 16% of data). In the ROI, the combination of low

CS and plentiful solar insolation provide the most favourable conditions for NPF days to occur, and on the contrary,

outside of the ROI, NPF days are much less likely to occur (i.e. bottom-right of Fig. 8b)). Moreover, the region

outside of the ROI with low solar insolation and low CS might be associated with increased cloudiness as the

incident radiation is reduced by the presence of clouds, and large accumulation mode particles are reduced due to

the effect of wet scavenging. The region with low solar insolation and high CS could be related to Arctic Haze

conditions.

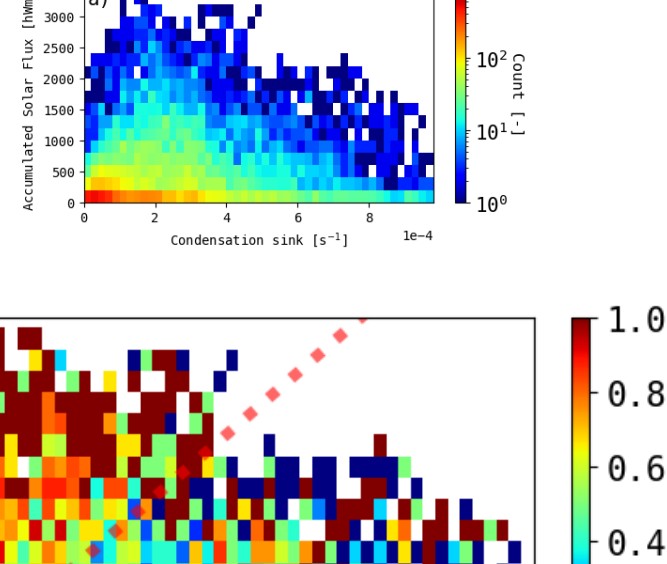

**Fig. 8: a) the total number of data points for the entire set of measurements (2022-04-17 to 2024-10-04) binned by the hourly accumulated solar insolation and condensation sink (CS) (CS between 0 - 10⁻³ s⁻¹ every 2·10⁻⁵ s⁻¹ and solar insolation between 0 - 4·10⁴ Wm⁻² every 200 Wm⁻²) and b) the likelihood, calculated by normalising the data set for the event days (defined using Aliaga et al. (2023)) by the total count for each given bin (i.e. a)). Likelihood is the fraction of measurement combinations lined to an NPF day The dashed red line defines the region of interest (ROI) whereby data on the left of it is considered to have the ideal conditions for NPF i.e. high accumulated solar flux and relatively low CS. The equation for the dashed red line is Accumulation solar Flux = 5·10⁶· CS.**

The ROI is defined indicatively using a simple straight line to separate ideal and non-ideal NPF conditions from each other. It should not be used as an empirical relationship, defining the occurrence of NPF. Instead it should be used as a guideline to highlight the general conditions most suitable for the promotion of NPF.

The data within the ROI and when they occur can be used to try and estimate the likelihood of a day experiencing an NPF event. A time series representing the daily likelihood of NPF occurrence (i.e. $P_{NPF,D}$) can be generated





from the ROI data (see Fig. 9) by counting the number of weekly data points within the ROI and dividing it by the total number of weekly data points either within or outside of the ROI (i.e. all). The weekly probabilities that

measurements fall within the ROI are interpolated to create the daily likelihood, $P_{NPF,D}$. $P_{NPF,D}$ generally, matches the frequency of NPF events as defined by Aliaga et al. (2023) or Dal Maso et al. (2005) (see Fig. 9). The spring and autumn peaks in $P_{NPF,D}$ and the event frequencies line up well through 2023, though in the summer (around July), the event frequency is high although daily $P_{NPF,D}$ is low. Using $P_{NPF,D}$ we can replicate the distinctive decline in the NPF frequency around July in 2022 and 2024. However, in July 2023, the decline in NPF events in

the observations is followed by a rapid increase which is not captured by $P_{NPF,D}$. Still, it is striking that the two parameters alone (CS and solar insolation) are able to serve as predictors for NPF probability, by representing the balance between the production and removal of NPF-precursors. The solar insolation reflects the photochemical production potential (likely via the production of hydroxyl radicals). However, the parameter combination does not account for the reactants required to produce the nucleating vapours (which may be $SO_2$ (likely via DMS)

and/or iodine compounds). Hence, deviations like those observed during July 2023 are expected as the conceptual model is partly incomplete due to a lack of adequate information regarding the original reactants. This said, however, the NPF frequency and likelihood largely seem limited by the balance between photochemical production potential and CS, and this in turn is suggestive of, to some degree, a self-regulating mechanism regarding the number of particles present in the nucleation and Aitken modes. There exist more sophisticated

models to predict NPF (Kuang et al., 2010).

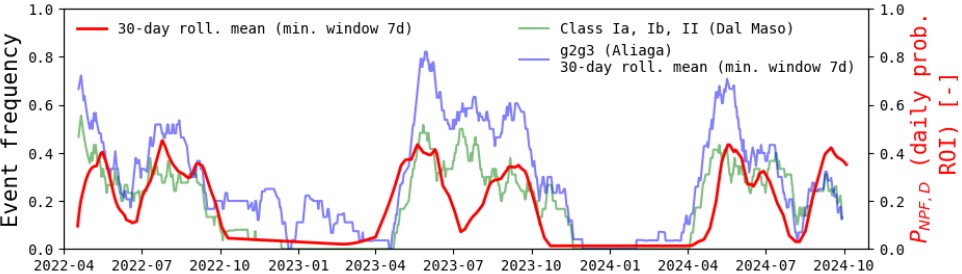

**Fig 9: The red line is $P_{NPF,D}$, the estimated daily likelihood of measurements occurring in the region of interest (ROI). The ROI is based on the relation between CS and 6-hour solar accumulation and is defined in figure 9b). The green and blue lines represent the rolling averages of the NPF daily event frequencies using the Dal Maso et al. (2005) and Aliaga et al. (2023) classification respectively. The green and blue curves are present to be able to compare $P_{NPF,D}$ with.**



**4.2 Dynamics driving NPF and growth**

During NPF events there is an excess availability of condensable precursors, e.g. SA, which contribute to the growth of newly formed and pre-existing particles, thus increasing CS. The increased CS enhances the removal

of condensable and nucleating vapours. As a result, a typical day with sustained NPF can lead to a relatively high CS ($>0.4\cdot10^{-3}$ s$^{-1}$), although the events tend to begin at low CS values (~$0.1\cdot10^{-3}$ s$^{-1}$) (see Fig. 10). It is worth noting, that during an event, not only do the nucleation mode particles grow in size, but also all other existing particles grow as well. Practically, this means that, the sink is doubled during the NPF-event. Consequently, and assuming steady state, this would mean that the source rate of the condensing species must double to sustain the

same concentration of condensing material.

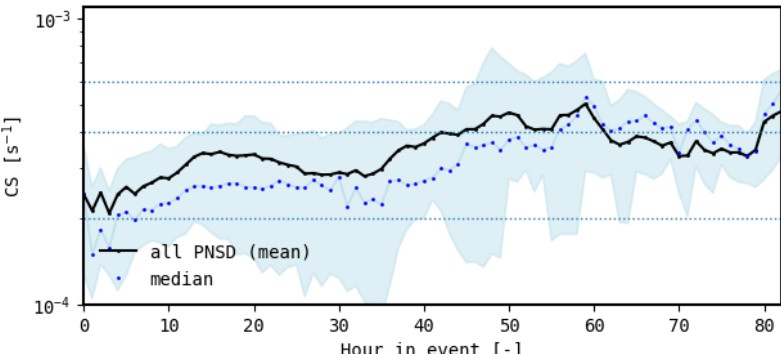

**Fig. 10: The calculated condensation sink (CS) during NPF events. It should be noted that the average NPF typically lasts for 10 hours; however, in some cases events can persist for days until the newly formed mode is removed via wet scavenging or advected away from ZEP. The black line is the mean and the blue is the median. The blue shaded region represents 25$^{th}$-75$^{th}$ percentiles. The dashed blue lines display CS values of 0.1, 0.2 and $0.4\cdot10^{-3}$ s$^{-1}$ and are there to guide the reader.**

This additional increase in CS during NPF events, shifts the ambient conditions away from the defined ROI ($>0.4\cdot10^{-3}$ s$^{-1}$) (see Fig. 8b)), thus potentially suppressing any subsequent NPF events. This mechanism will prevent further NPF, if the increase in CS during the NPF event is large enough to limit the availability of

precursors. This mechanism also helps to explain why the combination of low CS and high solar insolation measurements are seemly rare occurrences, as these conditions are soon followed by NPF, and increasing CS.

During the Arctic Haze period in late winter and early spring, high concentrations of accumulation mode aerosol dominate the contribution to the CS. Reduction in accumulation mode aerosol at the end of the Arctic Haze season



(i.e., end of May) should result in decreased CS, but ZEP continues to experience relatively high CS. We argue

that this relatively high sustained CS is the result of particle formation and growth within the Arctic; the increasing

frequency of NPF events during the summer months increases CS, and maintains the comparatively high CS

throughout the summer and the most intense part of the NPF season (see Fig. 2b)), additionally Tunved et al.

(2013)). Towards the late summer and autumn, the solar insolation has decreased substantially, and both marine

surface water productivity and photochemical production potential are declining. Together with low transport

efficiency from anthropogenic sources at lower latitudes and efficient wet removal, the aerosol reaches a minimum

with respect to CS and number during this part of the year.

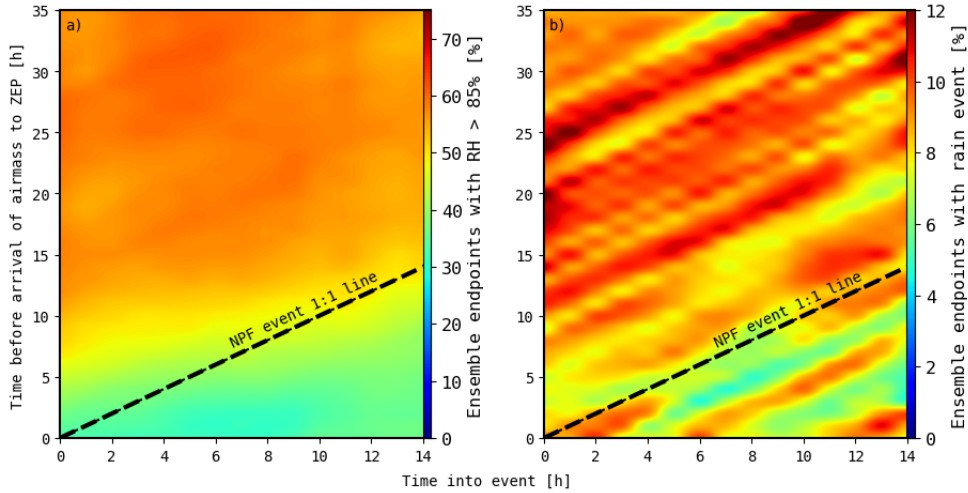

**Fig 11: Air mass history for a) percentage of back trajectory endpoints within an ensemble with a relative humidity above 85, and b) percentage of endpoints within ensemble which experienced a rainfall event (i.e. rainfall > 0mm). Y-axis is the number of hours backwards before the air masses arrive at ZEP, while the x-axis is the number of hours into the event e.g. if one event started at 6am and another at 9am, their start times would both be 0 hours. The black line is a 1:1 line which represents the start of the NPF event such that the grid points towards the top-left represent endpoints prior to the NPF event, whilst grid points towards the bottom-right represent endpoints during the NPF event. The x-axis has been sliced based on the typical duration of events (most events are <14 hours). This figure consists of the air mass history of Class Ia and Ib NPF events.**

In the Arctic environment, wet removal is a key factor responsible for the rapid reduction in CS (Garrett et al.,

2011).   For class Ia and Ib NPF events, we can observe that prior to the start an NPF event there is an increased

likelihood of cloudy conditions (RH >85%) and rainfall events (see Figs 11a-b). Figures 11a-b demonstrate how wet scavenging helps to effectively reduce the CS and provide more suitable conditions for NPF events to occur. The changes in the likelihood of cloudy conditions and rainfall prior to and post the start of an NPF event is slight, but significantly different, especially considering the duration prior to the event is longer and the likelihood of any rainfall should be compounded.

Any reduction in CS will shift the system towards the ROI, and as a result the system will rapidly respond to this reduction by producing new particles and grow the already existing ones. If suitable gaseous precursors and solar insolation are present, NPF will take place and the cycle will be reinitialised. This conceptual model, whilst highly simplified, offers substantial predictive value regarding the timing and frequency of NPF events (see Fig. 9). It does not, however, account for the original source of condensable and nucleating species nor does it consider cloud

activation and the in-cloud chemistry of Aitken mode particles in non-precipitating clouds.

### 4.3 NPF contribution to Aitken and potential CCN

Here, we argue that new particle formation is the main contributor to the Aitken mode in the Arctic; through frequent NPF events and the subsequent growth of nucleation mode particles, NPF activity is able to provide a significant proportion of the overall concentration of Aitken mode particles. We show in Sect 3.1.2 that at ZEP

the production of nucleation mode particles is largest at the end of spring, however, the production of newly formed particles persists throughout the summer months and into the autumn (see Fig. 2a)). During NPF events, including formation and subsequent growth the total number concentration of Aitken mode particles (5-70nm) is 3.6 times the amount compared with pre-event conditions. These events are relatively frequent (30 - 40% of summer days experience NPF, (see Fig. 1). Fundamentally, though, it is difficult to use in-situ measurements at a single

location to provide an estimation of the fraction of Aitken mode particles which are produced via NPF; for the Aitken mode particles at ZEP which are not directly linked to NPF activity, we cannot rule out the possibility that they are derived from NPF events outside of the vicinity of ZEP and then transported downwind.

At ZEP, NPF and the subsequent growth that follows, results in particles large enough to potentially be activated and become CCN; on numerous occasions (>50 times) we observed newly formed particles able to grow beyond

25nm while in the same regional event (see Fig. 12b), before being scavenged. Here, we use 25nm as the lowest bound for particles to act as CCN, however, it should be stated that this limit requires strong updrafts and extremely hygroscopic particles for particles to be activated. We observe that after ~20 hours of growth, we reach potential CCN-sizes at ZEP. It is important to note as well that once produced these particles most likely continue to grow



downstream of ZEP by condensation and in-cloud processing, so a large fraction of the Aitken mode should

inevitably reach CCN size in due time, as long as they are not otherwise scavenged.

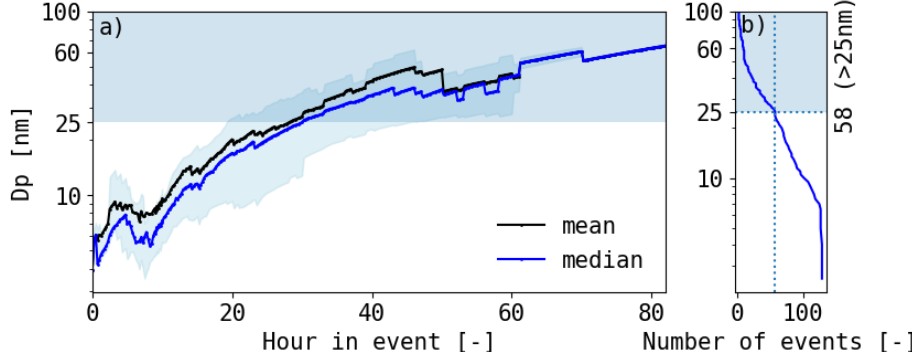

**Fig 12: Change in diameter during all class Ia, Ib and II NPF events. The blue shaded region covers 25nm – 60nm.**

**(b) displays the number of NPF events against the maximum diameter reached during growth. The horizontal**

**dashed line marks 25nm whilst the vertical dashed line marks the number of NPF events which reached 25nm.**

It is difficult to ascertain the proportion of NPF-derived Aitken mode particles that actually become CCN before

being scavenged or deposited, and estimating an exact fraction is beyond the scope of this study. However,

previous modelling and measurement studies can help to shed some light on the likelihood that NPF events as

measured from ZEP can contribution to the CCN budget. Jung et al. (2018) demonstrate that at supersaturations

(ss) greater than 0.4% there is a considerable number of CCN measured during the summer months at ZEP.

Moreover, the changes in sources of CCN-sized aerosol particles, from mainly anthropogenic to NPF-derived,

during the Haze to summer transition can be seen in the reduction of the geometric mean diameter. For summer,

Jung et al. (2018) suggest a significant number of particles >10nm are activated into cloud droplets, as the

activation ratio during summer is around 0.5 for fairly high ss, i.e. >0.6%. In addition, Karlsson et al. (2021)

show that Aitken mode-sized cloud residuals are fairly common in the summer time (i.e. July onwards); the

transition from accumulation mode-dominated to Aitken mode-dominated aerosol did not lead to a significant

reduction in the concentration of cloud residuals, indicating that particles as small as 20nm can be activated. Motos

et al. (2023) suggest that particles >20nm are activated in the summer, with a modal activation diameter of 60nm

showing that Aitken mode particles act as CCN.



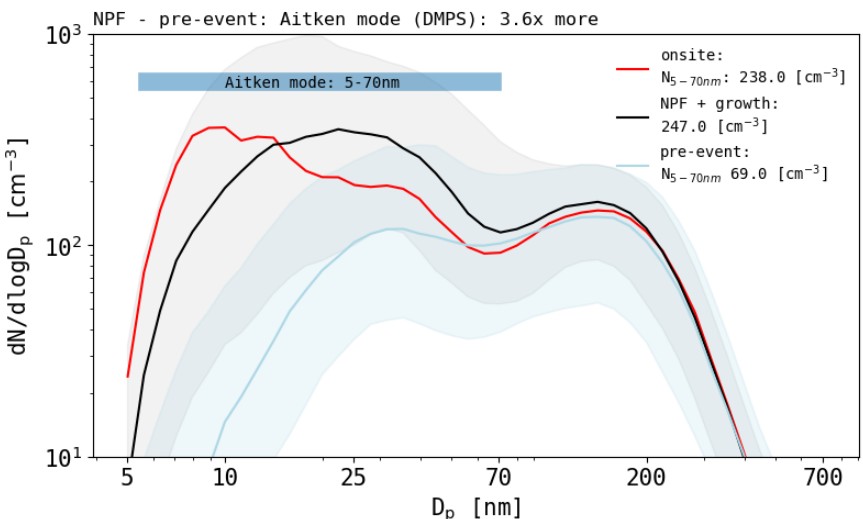

**Fig 13: The particle number size distributions (DMPS) for periods before the NPF events (1-2 hours before event started) (pre-event, light blue), during NPF and growth (black) and on-site nucleation (production of 2-4nm particles, red, *on-site formation*).**

**5. Conclusion**

In this study, we explore the annual cycle of atmospheric new particle formation (NPF) events from almost 3-years' worth of measurements (2022-2024). In keeping with previous studies, e.g., Tunved et al. (2013) and Ström et al. (2009), the Arctic summer and more specifically the end of spring and start of summer represent a peak in

NPF intensity as observed at Zeppelin Observatory (ZEP). We find that this peak in the strength of NPF events coincides roughly with the maximum solar insolation (defined as the last 6 hours of accumulated solar radiation along HYSPLIT back trajectories within the mixed-layer arriving at ZEP); solar insolation allows for the increased availability of precursors by encouraging oxidation. The NPF frequency during the summer remains fairly consistent, with an estimated 20-30% of days experiencing an NPF event. Towards the end of summer, the decline

in the amount of solar insolation reduces the NPF intensity. In addition, it is also apparent that the condensation sink (CS i.e. linked to the total surface area of pre-existing aerosol) plays an important role, as the NPF intensity is more pronounced when there are reductions in CS. NPF events typically begin when there is less of a sink, in regard to precursors.

Typically, as the Arctic transitions from the Haze season to the summertime cloud coverage and precipitation

increase and this in turn reduces the once predominant accumulation mode via wet scavenging. However, the increased cloud cover, not only reduces the CS, but also reduces the solar insolation. Hence, the transition to the

summertime both promotes and discourages NPF. These competing effects can explain why we observe such a reduction in NPF intensity as the summer progresses towards the end. We can only speculate that a warming climate may appear more similar to the late Arctic summer, being both warmer and wetter. Some studies indicate

that cloud cover in the Arctic could potentially increase (Barton and Veron, 2012) or has (Eastman and Warren, 2010; Francis et al., 2009; Kay and Gettelman, 2009; Palm et al., 2010; Vihma et al., 2008) with periods of repaid sea ice loss.

One main finding from this study is that we can, with rather good accuracy, predict the occurrence of an NPF day (Fig. 9) by using CS and solar insolation as the two main predictive parameters. Periods exhibiting low CS and

high solar insolation provide the most ideal conditions for NPF events and the occurrence of these periods can reproduce the observed frequency of NPF events. It should be noted, though, that other factors play a role. The sources to the precursors of the condensable vapours, as well as additional meteorological and environmental parameters such as sea-ice cover, chlorophyll a concentration and the varying contributions from different source regions all can influence the occurrence and strength of NPF. However, from this study we argue that both solar

insolation and CS can be used to create a rather simple predictive model. The Artic perhaps offers a unique environment, where the relationship between solar insolation and NPF is potentially stronger, due to the typically lower CS and lower overall solar radiation intensity (Bousiotis et al., 2021).

Quantifying the contribution of CCN from NPF on low-level clouds is beyond the scope of this study. However, we are able to demonstrate that periods of formation and subsequent growth lead to a significant increase in the

proportion of the total number of Aitken mode particles. In combination with this, we show that on many occasions during NPF events, the nucleation mode particles grow to potential CCN sizes once they are formed (>50 NPF events lead to diameters >25nm). With the correct conditions, i.e. a relatively strong updraft, the formation of particles >25nm could lead to droplet formation and in turn alter cloud properties.

Given the high frequency of onsite NPF and the large potential area for formation, it is not unlikely that a majority

of Aitken mode particles in fact originate from NPF within or close to the Arctic region. Provided sufficient time for growth up to CCN-sizes, it seems clear that NPF events constitute a significant source of CCN in the Arctic. For a more detailed picture of the aerosol budget in the Arctic and a more quantitative account of the NPF-contribution to the overall Arctic CCN-population, large scale modelling evaluated against observations and combined with process studies are required.


**Code availability**

Code will be made available on GitHub when published.

**Data availability**

Data will be on the Bolin Centre database when published (https://bolin.su.se/data/).

**Author Contributions**

Dominic Heslin-Rees (DHR), Radovan Krejci (RK), and Peter Tunved (PT) worked together to develop the research questions and design the study. DHR performed data analysis and wrote the paper together with assistance from RK and PT. There was good input and advice along the way from Ilona Riipinen (IR), Annica Ekman (AK), Diego Aliaga (DA) and Martina Mazzini (MMi). Janne Lampilahti (JL) developed many of the NAIS specific Python packages and showed DHR how to clean and service the NAIS. Stefania Gilardoni (SG) provided comments. SG, Mikko Sipilä (MS), Mauro Mazzola (MMa) and Roseline Thakur (RT) helped with the Gruvabadet work. Ki-Tae Park (KTP) and Kitack Lee (KT) maintained and operated the Zeppelin DMS analyzer are part of a collaborative effort between KOPRI and POSTECH research team. Kihong Park (KP) and Young Jun Yoon (YJY) provided the Nano-SMPS data.

**Acknowledgements**

This study would not be possible without the work done by the research engineers Birgitta Noone, Tabea Henning, and Ondrej Tesar from ACES and the staff from the Norwegian Polar Institute (NPI). NPI provide invaluable on-site support including attending to the NAIS. NPI is also acknowledged for substantial long-term support in maintaining the measurements at Zeppelin Observatory (ZEP). We would also like to thank Kai Rosman for developing the software we used for the instrumentation at ZEP. We would like to thank Ove Hermansen from the Norwegian Institute for Air Research (NILU) for providing the ambient meteorological data.

**Financial Support**

Aerosol size distribution measurements are supported by the Swedish EPA. Swedish participation was supported by Swedish Environmental Protection Agency (Naturvårdsverket), Knut and Allice Wallenberg foundation (KWA) and FORMAS funding agency.

Knut-and-Alice-Wallenberg Foundation within the Arctic Climate Across Scales (Project No. \,2016.0024), the Swedish EPA's (Naturvårdsverket) Environmental monitoring program (Miljöövervakning), the Swedish



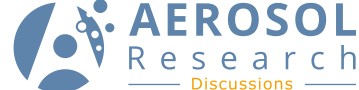

Research Council FORMAS (Project "Interplay between water, clouds and Aerosols in the Arctic," \# 2016-01427)

The aerosol observations at the Zeppelin station have been supported by the KAW Stiftelse (grant no. 2016.0024), the Swedish Environmental Protection agency (Naturvårdsverket), and the ACTRIS Sweden project supported by the Swedish Research Council

The KAW project CLIVE and VR financed infrastructure project ACTRIS-Sweden.




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
