# Peer review of "Drivers governing the seasonality of new particle formation in"

_Aerosol Research, 2025_

## Referee Comment (RC1)

Review of "Drivers governing the seasonality of new particle formation in the Arctic" by Heslin-Rees et al.

This paper gives an overview of new particle formation (NPF) events at Zeppelin Observatory on Svalbard in the High Arctic. It gives a comprehensive analysis of the seasonal frequency, growth and formation rates, duration and starting times, air mass origin, the factors driving NPF, and their contribution to Aitken mode particles and their ability to grow to cloud condensation nuclei (CCN) sizes. NPF events during polar night are also shown, something that is rarely observed. The authors argue that accumulated solar radiation in the past 6 hours (termed solar insolation) and the condensation sink (CS), measured locally, are the driving factors for NPF. They developed a simplified predictive model using these two parameters to estimate the likelihood of observing NPF. This model is compared to the Dal Maso NPF classification and Aliaga Nanoparticle Ranking methods for analyzing NPF activity. The authors do an excellent job of presenting the code sources throughout the text. This is the first evaluation of the Aliaga method in the Arctic. This paper fits within the scope of AR and will be of interest to the atmospheric community. This paper describes several interesting aspects of NPF and will undoubtedly add to the knowledge base of Arctic NPF, however there are some concerns that need to be addressed. Overall, I would recommend publication after major corrections.

Main Concerns:

My main concern is the lack of quantitative statistics utilized in this study, the use of rolling means with large window sizes, and comparing results visually. Two weeks and one month window sizes are very large and coupled with a logarithmic scale can make visually interpreting the results difficult. I would encourage the authors to include more quantitative statistical methods on unsmoothed or minimal smoothing the data to explore the main drivers of NPF and evaluate their simplified predictive model. To give some suggestions, this could include non-parametric methods such as Spearman rank correlations, binning variables and comparing differences between these groups using the Student's t, Welsh's t, or Mann-Whiteny U test (depending on the normality and homogeneity of variance of the groups) for the drivers of NPF and more classical methods such as coefficient of determination, (root) mean square error, and mean absolute error for the evaluation of the simplified predictive model, although the exact methods are up to the authors.

I would agree with the author's interpretation that solar insolation and CS are important factors driving NPF, for the same reasons the authors nicely described, but the authors do not quantitatively demonstrate this in their manuscript. The authors should provide more convincing quantitative evidence in their manuscript.

The same can be said for their simplified predictive model, visually it agrees with the two other classifications schemes although there are notable differences in the highly smoothed time series. The authors state that April 2022 is limited in sample size which could explain this disagreement. The Aliaga method shows non-zero frequency during winter of 2022/23, I am wondering if all the blowing snow events were removed during this time, although I doubt such a large period would be missed. While I understand that a thorough evaluation of the Aliaga method is not the main purpose of this manuscript, can the authors comment on why the Aliaga method gives a non-zero frequency during the winter of 2022/2023 as opposed to the following winter? During July 2023, the simplified predictive model shows disagreement with the other methods, the authors state that this is expected to be due to the lack of adequate information regarding original reactants, do the authors have any evidence to support this expectation? There are other factors that could affect this disagreement, such as meteorology, air mass origin in relation to biologically active and sea ice regions, and wet removal. Have the authors explored other factors that could influence the observed disagreement during these periods?

Finally, the daily likelihood of NPF occurrence ($P_{NPF,D}$) is actually calculated on a weekly basis then interpolated (how?) to a daily resolution. This interpolated timeseries is then smoothed with a 30-day rolling mean. Why was the daily likelihood of NPF occurrence not calculated on a daily basis as the name implies? I would be interested in how the $P_{NPF,D}$ compares, using quantitative statistics, to the other two methods. A quantitative comparison could also be made between the Dal Maso and Aliaga methods, as this is the first time the Aliaga method has been utilized in the Arctic (to the reviewer's knowledge) and this could provide a valuable assessment of its performance in a polar environment for future studies. Although it is understandable if this assessment is deemed outside of the scope of the article.

Minor concerns:

Considering solar radiation along the back trajectory is an excellent idea as this would give a better indication of photochemical activity compared to in situ solar radiation and allows for appropriate time for oxidation of precursor gases to occur although I am now wondering if the authors considered something similar for the CS as well. Obviously, HYSPLIT cannot give an indication of the CS along the back trajectory but have the authors considered lagging the CS or possibly using accumulated precipitation along the back trajectory? They demonstrate precipitation and cloud processing has a large effect on NPF in Fig 11 and state this on 683. Tunved et al. (2013) and Khadir et al. (2023) also demonstrate the effect of accumulated precipitation on ultrafine aerosol particles nicely from ZEP. The authors demonstrate the NPF events will increase the CS, so I am

wondering if the authors explored other variables, such as accumulated precipitation, to include in the simplified predictive model.

For the simplified predictive model, how did the authors calculate the line defining the ROI? Was it based on visual inspection or adjusted based on a criterion? I am very curious as to how did the authors calculate these averages "mean (median) 66% (68.4%)"? Did they simply take the mean and median of the ratios in the ROI? The large differences in the number of observations in different regions of the ROI could make such averaging give misleading results. Were the ratios weighed by the number of observations in each grid cell, i.e., weighted mean or median? I am wondering if a weighted average or calculating the sum the numerators and denominators of each grid cell in the ROI (i.e., integrating over the ROI) would be a more robust statistic. Could the authors please clarify this?

The ROI does have a higher likelihood of observing an NPF day compared to outside. However, there are grid cells with high likelihoods outside of the ROI although these grid cells do have a small sample size. I am wondering if the authors could comment on these false negatives in their model.

Line by line comments:

Line 49: Maybe it is worth mentioning ZEP is a mountain site when describing the geographic location.

Line 55: "which matches the frequency" Here would be a good place to include the performance evaluation for their model.

Line 57: I would say the authors nicely demonstrate the polar night NPF events arrive from higher altitudes (Fig. S17) therefore they are not speculating.

Lines 61-63: The two parts of this sentence are redundant. Please make this sentence more concise.

Line 63: "over 50 NPF events" Would this be more accurately expressed as a percentage of total NPF events for context? This would help contextualize the importance of NPF's contribution to CCN in the Arctic.

Line 90-92: Could the authors include more original references for this statement? Also, Carslaw, 2022 is not in the reference list.

Lines 109-111: The authors mention both negative and positive ions but only give a reference for negative ions, can the authors include a reference for positive ions?

Line 122: Villum Research Station is the preferred name for this station.

Line 125: The town of Barrow, Alaska was renamed to its original, indigenous name of Utqiaġvik in 2016. However, the NOAA observatory located outside of the town is still

referred to as the Barrow Atmospheric Baseline Observatory (BRW). I would encourage the authors to either refer to the town by its proper name or be explicit when referring to the atmospheric observatory.

Lines 128-131: Beck et al. (2021) also showed different species contributing to NPF and growth during different times of the year at these two locations, iodic acid at Villum during springtime and sulfuric acid/ammonium during summer; sulfuric acid/ammonium at ZEP during springtime and highly oxygenated molecules during summer. Could the authors please mention these important results from Beck et al. (2021)?

Line 132: Please define DMS and mention its marine, biogenic origins. The authors mention phytoplankton blooms on lines 86-87, here would be a good place to first mention DMS and its effect on NPF.

Lines 132-135: As currently constructed, this sentence implies the correlation between DMS and chlorophyll a is the reason that DMS is an important source of nucleating vapors. While none of the information in this sentence is incorrect, the way the sentence is structured is misleading, please rephrase.

Line 146: "higher end of the accumulation mode" do the authors mean Aitken mode here?

Line 176: Are two CPCs in this sentence referring to the ones on the twin DMPS system? Or the two CPCs references in the next sentence? If referring to the DMPS system, then saying "total aerosol concentration" for size segregated monodispersed aerosol is confusing.

Line 244: Would it be possible to show a figure of exemplary days for Class Ia, Ib, and II NPF event in the supplement? Given there is a degree of subjectivity in the Dal Maso method, this would help readers who are unfamiliar with the method gain context for each event.

Line 283: Was the growth rate calculated using the timestamp for the maximum rate of change for each size bin as listed in the text or the maximum concentration as listed on line 301?

Equation 1: $N_{i-j}$ is not defined.

Line 300-301: The link does not work and there is no repository named npf_event_analyzer. on jlpl's Github page.

Equation 3: CS is not in Eq. 3 but is listed in the description. Is it part of another term in the equation?

Line 320: Was the starting height above ground level or mean sea level?

Sects. 2.4: I commend the authors on calculating ensemble trajectories as this gives a more robust estimate of air mass source regions compared to a single back trajectory.

They are also a bit more complex to work with. I am curious if the authors combined all ensemble members or used individual ensemble members for the calculation of chlorophyll a exposure, nucleation site estimation, and air mass source regions. Could the authors please explain this in the text?

Sect. 2.4.2: Could the equation for calculating chlorophyll a exposure be listed here? Readers might not be familiar with the work of Park et al. (2018). I am also curious as to why the authors calculated chlorophyll a exposure for 2.5 years of trajectories (no easy task) and do not show any of the results even though they are mentioned several times throughout the text.

Sect. 3.1.1: "Occurrence" and "frequency" are used interchangeably, I am wondering if it is better to be consistent to avoid confusion.

Lines 379-383: The authors do an excellent job of describing previous studies on NPF activity at ZEP, could the authors include studies from other Arctic sites to provide context.

Line 390: "maximum daily increase" should this be "maximum daily concentration" as listed on line 266?

Lines 400-401 and 407-408: This is where some quantitative statistical metrics could be included.

Line 411-412: What other precursor candidates are there for sulfuric acid production other than $SO_2$?

Lines 412-413: The CS is for condensing vapors and the CoagS is for coagulating particles, I think these terms are switched in these lines.

Lines 417-418: Can the authors simply list the changes in cloud cover or their frequencies from Maturilli and Ebell (2018)?

Lines 423-427: It is interesting to note the almost anti-correlation between DMS and $\Delta N_{max\ 2.82\text{-}5}$.

Lines 434-436: Would the polar night events be better presented as a table? What is the meaning of 3ab and x2 in these lines?

Lines 438-439: The latitudinal dependency is explained but was there any longitudinal dependency of these polar night events?

Lines 439-441: Wouldn't the low temperatures experienced by high altitude air masses during this time of year also lower the volatility of any nucleating vapors, making the process more efficient?

Line 441: Utqiaġvik is misspelled.

Line 465: Could the authors mention the differences in the definition of the onset of nucleation between these studies?

Sect. 3.1.5: The section is titled "Seasonality of Growth Rates and formation rates" yet no seasonality of formation rates is presented. I would suggest adding a subpanel to Fig. 4 which shows the seasonality of formation rates and mentioning the seasonality in the text. There should be a space between nm and $hr^{-1}$, also in the figure axis label. What sites were included in Kerminen et al. (2018) for the Arctic?

Sect. 3.1.6: I am curious why the authors did not compare Cv to the results from Beck et al. (2021) as well. Observations from MOSAiC are highly valuable in this context but Beck et al. was from Gruvebadet at the base of Mount Zeppelin. The MOSAiC year was also shown to be anomalous in terms of the Arctic Oscillation and transport from Eurasia during the winter months (Boyer et al., 2023). The caption of Fig. 5 reads as if the measurements from Boyer et al. cover the years 2022-2024.

Lines 510-511: Could the other species possibly be HOMs? Which were reported by Beck et al. (2021) but not Boyer et al. (2024).

Header of Table 1: The unit for CS needs a superscript. The unit for solar insolation is given as $hW\ m^{-2}$ elsewhere, also should there not be a space between hW and $m^{-2}$?

Caption and label of Fig. 7: I am wondering the term "Nucleation site" is confusing and would it be better to express it more generally as "Count per grid box" as expressed in previous publications (Heslin-Rees et al., 2024). Is the NAIS defining the nucleation sites or is it HYSPLIT?

Lines 575-578: I would encourage the authors to include their results here as this is an interesting finding that chlorophyll a exposure is not correlated to DMS.

Lines 578-583: Can the authors give examples of "various other meteorological and environmental factors"? Also, could there any methodological factors influencing the lack of correlation between chlorophyll a exposure and DMS, such as inaccuracies in HYSPLIT and satellite retrievals of chlorophyll a due to sea ice and cloud cover?

Lines 585-588: I would suggest removing this sentence as it does not add to the discussion nor explain their results, although this is only my opinion.

Lines 589-603: The authors do a nice job of explaining the differences between the Greenland Sea and the Barents Sea for NPF source regions, although another region highlighted in Fig. 7 is the Arctic Ocean. Could emissions from sea ice, e.g. iodine and sympagic algae be contributing to this observation? This might be worth mentioning.

Lines 606-608: Here is an example of where quantitative statistics could be used to support the results. For instance, the distributions of solar insolation on NPF vs non-NPF event days could be compared or the anomaly of the CS to the seasonal average could

be analyzed for NPF event days. Such an analysis would make their justification of the simplified predictive model more robust and convincing.

Lines 613-622: The units for solar insolation are sometimes given as hWm$^{-2}$ and sometimes as Wm$^{-2}$, additionally there should be a space between W and m$^{-2}$ otherwise it reads as if both units are in the denominator. Units for CS are missing.

Line 617: I am curious how this relationship is for the Dal Maso method, does it still hold?

Fig 8: I would suggest making the two subpanels the same size and please use a color-blind friendly color bar.

Lines 627: Could the authors quantify how much less likely NPF events are outside of the ROI?

Line 693: Another example of where quantitative statistics would help as this "substantial predictive value" is not demonstrated.

Line 701-703: Figure 13 is not referenced in the text. Should it be referenced in this sentence?

Line 709: Why not be precise with an exact number of NPF events that grew over 25nm?

Line 710: In the Arctic summertime, the lack of accumulation mode particles can drive the supersaturation to high values, activating small particles as the authors detail nicely, thus 25 nm is a lower limit. Could the authors give the number of NPF events that surpass other thresholds for activation, such as 60, 70, or 80 nm (the exact thresholds are left up to the authors). This would give a range of the number of NPF events that could contribute to CCN.

Line 712-713: Figure 12a is not referenced in the text, should it be referenced in this sentence? What instruments were used to calculate the mean and median diameter and how were these calculated? Using the mean and median, it appears that the diameter reaches 25 nm after approx. 30 hours. The shading on the lines (which is not mentioned in the figure caption) appears to reach 25nm after 20 hours.

Line 753: This accuracy is not quantified.

Supplement:

Fig. S1 caption: Could the authors list the names of all the parameters in paratheses? The authors state 5 tubing parameters but more than 5 are given.

Fig. S2a: The legend significantly overlaps with the data and the abbreviations in the legend are not given.

Fig. S6 caption: What is the meaning of "These are overlapping – improvement representation"?

S2.5: The authors state "There are numerous papers which highlight this common discrepancy." but none are cited, please cites numerous papers to support this statement.

S2.5 below Fig. S9: Fig. X? Should this be Fig. S11?

Fig. S10: The caption does not explain the figure at all. Please rewrite the caption to accurately describe the figure. Maybe this text was meant to be included elsewhere?

Y axis of Figs. S10 and S11: What is dndlDp? Should this be dN/dlogDp as used elsewhere in the manuscript? From the caption, I gathered this is the slope of the regression line, but "Coef." is listed in the label. Finally, "(DMPS/NAIS-5)" reads as if 5 was subtracted from the ratio of the DMPS to the NAIS, although it was clear from Sect. 2.2 that it is NAIS5. Please adjust the y axis label to be clearer.

Fig. S12: There is no (b) panel in the figure. Is "N int max" in the title supposed to indicate $\Delta N_{max, 2.8-5}$?

Fig. S13: What is dNint,max and d_N_int_max? These terms are not defined. Should the ratio for the mapping of g1, g2, and g3 be 1:1:1 since there are three groups but only two numbers in the ratio?

Fig. S16: Should the units for the y label be in nm? Also, no color bar is given.

Fig. S17: What are the vertical dashed lines in this figure?

Figs. S2, S7, S8, S16 present a lot of information, therefore I would suggest making these figures larger, possibly the width of the page. This would make it easier for the reader to grasp all the details presented in these nice figures.

A bibliography is missing from the supplement.

References

Beck, L. J., Sarnela, N., Junninen, H., Hoppe, C. J. M., Garmash, O., Bianchi, F., Riva, M., Rose, C., Peräkylä, O., Wimmer, D., Kausiala, O., Jokinen, T., Ahonen, L., Mikkilä, J., Hakala, J., He, X.-C., Kontkanen, J., Wolf, K. K. E., Cappelletti, D., Mazzola, M., Traversi, R., Petroselli, C., Viola, A. P., Vitale, V., Lange, R., Massling, A., Nøjgaard, J. K., Krejci, R., Karlsson, L., Zieger, P., Jang, S., Lee, K., Vakkari, V., Lampilahti, J., Thakur, R. C., Leino, K., Kangasluoma, J., Duplissy, E.-M., Siivola, E., Marbouti, M., Tham, Y. J., Saiz-Lopez, A., Petäjä, T., Ehn, M., Worsnop, D. R., Skov, H., Kulmala, M., Kerminen, V.-M., and Sipilä, M.: Differing Mechanisms of New Particle Formation at Two Arctic Sites, Geophys. Res. Lett., 48, e2020GL091334, https://doi.org/10.1029/2020GL091334, 2021.

Boyer, M., Aliaga, D., Pernov, J. B., Angot, H., Quéléver, L. L. J., Dada, L., Heutte, B., Dall'Osto, M., Beddows, D. C. S., Brasseur, Z., Beck, I., Bucci, S., Duetsch, M., Stohl, A., Laurila, T., Asmi, E., Massling, A., Thomas, D. C., Nøjgaard, J. K., Chan, T., Sharma, S., Tunved, P., Krejci, R., Hansson, H. C., Bianchi, F., Lehtipalo, K., Wiedensohler, A., Weinhold, K., Kulmala, M., Petäjä, T., Sipilä, M., Schmale, J., and Jokinen, T.: A full year of aerosol size distribution data from the central Arctic under an extreme positive Arctic Oscillation: insights from the Multidisciplinary drifting Observatory for the Study of Arctic Climate (MOSAiC) expedition, Atmospheric Chem. Phys., 23, 389–415, https://doi.org/10.5194/acp-23-389-2023, 2023.

Boyer, M., Aliaga, D., Quéléver, L. L. J., Bucci, S., Angot, H., Dada, L., Heutte, B., Beck, L., Duetsch, M., Stohl, A., Beck, I., Laurila, T., Sarnela, N., Thakur, R. C., Miljevic, B., Kulmala, M., Petäjä, T., Sipilä, M., Schmale, J., and Jokinen, T.: The annual cycle and sources of relevant aerosol precursor vapors in the central Arctic during the MOSAiC expedition, Atmospheric Chem. Phys., 24, 12595–12621, https://doi.org/10.5194/acp-24-12595-2024, 2024.

Heslin-Rees, D., Tunved, P., Ström, J., Cremer, R., Zieger, P., Riipinen, I., Ekman, A. M. L., Eleftheriadis, K., and Krejci, R.: Increase in precipitation scavenging contributes to long-term reductions of light-absorbing aerosol in the Arctic, Atmospheric Chem. Phys., 24, 2059–2075, https://doi.org/10.5194/acp-24-2059-2024, 2024.

Kerminen, V.-M., Chen, X., Vakkari, V., Petäjä, T., Kulmala, M., and Bianchi, F.: Atmospheric new particle formation and growth: review of field observations, Environ. Res. Lett., 13, https://doi.org/10.1088/1748-9326/aadf3c, 2018.

Khadir, T., Riipinen, I., Talvinen, S., Heslin-Rees, D., Pöhlker, C., Rizzo, L., Machado, L. A. T., Franco, M. A., Kremper, L. A., Artaxo, P., Petäjä, T., Kulmala, M., Tunved, P., Ekman, A. M. L., Krejci, R., and Virtanen, A.: Sink, Source or Something In-Between? Net Effects of Precipitation on Aerosol Particle Populations, Geophys. Res. Lett., 50, e2023GL104325, https://doi.org/10.1029/2023GL104325, 2023.

Maturilli, M. and Ebell, K.: Twenty-five years of cloud base height measurements by ceilometer in Ny-Ålesund, Svalbard, Earth Syst. Sci. Data, 10, 1451–1456, https://doi.org/10.5194/essd-10-1451-2018, 2018.

Park, K.-T., Lee, K., Kim, T.-W., Yoon, Y. J., Jang, E.-H., Jang, S., Lee, B.-Y., and Hermansen, O.: Atmospheric DMS in the Arctic Ocean and Its Relation to Phytoplankton Biomass, Glob. Biogeochem. Cycles, 32, 351–359, https://doi.org/10.1002/2017GB005805, 2018.

Tunved, P., Ström, J., and Krejci, R.: Arctic aerosol life cycle: linking aerosol size distributions observed between 2000 and 2010 with air mass transport and precipitation at Zeppelin station, Ny-Ålesund, Svalbard, Atmospheric Chem. Phys., 13, 3643–3660, https://doi.org/10.5194/acp-13-3643-2013, 2013.

---

## Author Comment (AC1)

**Reply to reviewers of the manuscript "Drivers governing the seasonality of new particle formation in the Arctic"**

Heslin-Rees et al.

October 2025

We thank both reviewers for their positive and constructive comments. We have modified our manuscript based on their suggestions. Please find our detailed reply below (given in blue colour).

**1 Reviewer 1**

This paper gives an overview of new particle formation (NPF) events at Zeppelin Observatory on Svalbard in the High Arctic. It gives a comprehensive analysis of the seasonal frequency, growth and formation rates, duration and starting times, air mass origin, the factors driving NPF, and their contribution to Aitken mode particles and their ability to grow to cloud condensation nuclei (CCN) sizes. NPF events during polar night are also shown, something that is rarely observed. The authors argue that accumulated solar radiation in the past 6 hours (termed solar insolation) and the condensation sink (CS), measured locally, are the driving factors for NPF. They developed a simplified predictive model using these two parameters to estimate the likelihood of observing NPF. This model is compared to the Dal Maso NPF classification and Aliaga Nanoparticle Ranking methods for analyzing NPF activity. The authors do an excellent job of presenting the code sources throughout the text. This is the first evaluation of the Aliaga method in the Arctic. This paper fits within the scope of AR and will be of interest to the atmospheric community. This paper describes several interesting aspects of NPF and will undoubtedly add to the knowledge base of Arctic NPF, however there are some concerns that need to be addressed. Overall, I would recommend publication after major corrections.

**Main Concerns:**

My main concern is the lack of quantitative statistics utilized in this study, the use of rolling means with large window sizes, and comparing results visually. Two weeks and one month window sizes are very large and coupled with a logarithmic scale can make visually interpreting the results difficult. I would encourage the authors to include more quantitative statistical methods on unsmoothed or minimal smoothing the data to explore the main drivers of NPF and evaluate their simplified predictive model. To give some suggestions, this could include non-parametric methods such as Spearman rank correlations, binning variables and comparing differences between these groups using the Student's t, Welsh's t, or Mann-Whiteny U test (depending on the normality and homogeneity of variance of the groups) for the drivers of NPF and more classical methods such as coefficient of

determination, (root) mean square error, and mean absolute error for the evaluation of the simplified predictive model, although the exact methods are up to the authors.

I would agree with the author's interpretation that solar insolation and CS are important factors driving NPF, for the same reasons the authors nicely described, but the authors do not quantitatively demonstrate this in their manuscript. The authors should provide more convincing quantitative evidence in their manuscript. The same can be said for their simplified predictive model, visually it agrees with the two other classifications schemes although there are notable differences in the highly smoothed time series. The authors state that April 2022 is limited in sample size which could explain this disagreement. The Aliaga method shows non-zero frequency during winter of 2022/23, I am wondering if all the blowing snow events were removed during this time, although I doubt such a large period would be missed. While I understand that a thorough evaluation of the Aliaga method is not the main purpose of this manuscript, can the authors comment on why the Aliaga method gives a non-zero frequency during the winter of 2022/2023 as opposed to the following winter? During July 2023, the simplified predictive model shows disagreement with the other methods, the authors state that this is expected to be due to the lack of adequate information regarding original reactants, do the authors have any evidence to support this expectation? There are other factors that could affect this disagreement, such as meteorology, air mass origin in relation to biologically active and sea ice regions, and wet removal. Have the authors explored other factors that could influence the observed disagreement during these periods?

Finally, the daily likelihood of NPF occurrence (PNPF,D) is actually calculated on a weekly basis then interpolated (how?) to a daily resolution. This interpolated timeseries is then smoothed with a 30-day rolling mean. Why was the daily likelihood of NPF occurrence not calculated on a daily basis as the name implies? I would be interested in how the PNPF,D compares, using quantitative statistics, to the other two methods. A quantitative comparison could also be made between the Dal Maso and Aliaga methods, as this is the first time the Aliaga method has been utilized in the Arctic (to the reviewer's knowledge) and this could provide a valuable assessment of its performance in a polar environment for future studies. Although it is understandable if this assessment is deemed outside of the scope of the article.

**Response to main concerns:**

**Lack of quantitative statistics:**

In the paper, we have now used the Mann-Whitney U test to demonstrate how the ROI region is not identical to the non-ROI region in terms of NPF probability/likelihood. Furthermore, we have used the ordinary least squares method to examine how the predictive model compares to the frequency of NPF events. Finally, we explored how applying a rolling average of varying-size windows changed the coefficient of determination for the predictive model and observation comparison.

**Rolling means:**

The reviewer comments that two weeks and or one month size windows are very large, we have take the approach that we would like to observe how NPF through a roughly "synoptic scale" i.e., weather phenomena occurring on a large spatial scale with a timescale of days to weeks.

The exact amount of smoothing that is used to try and tease out the seasonality of NPF and related parameters is of a subjective nature. Deciding to use 4-days, 7-days, 14-days or 30-days depends on the reader. When possible we have added figures which detail the change when the

length of the rolling average is changed. For some figures though, a significant rolling mean was required as we were dealing with binary values (i.e., in the case of NPF frequency).

In Figure 1 we attempted to inspect the characteristic number of peaks which represent the NPF season. Obviously, the number of peaks increases with increased average windows. However, we believe that rolling averaging windows of greater 10 allow for the characteristics peaks to show.

Figure 1: Different rolling mean windows applied to the event frequencies, and to the intensity parameter.

**Mann-Whiteny U test:**

The non-parameteric Mann-Whiteny U test was utilised to test the significance of our so called region of interest (ROI). The reason for this was to show to the reader that this region really does stand out given the combination of solar insolation and CS in driving NPF. Furthermore, we used the Mann-Whiteny U test to show how CS and Solar insolation were different depending on if event or non-event periods were chosen.

**Predictive model:**

Instead of using the frequency of ROI as our predictive test, we decided to opp for SUNFLUX/CS, and compare it to the event frequency as defined by the Nano-ranking classification scheme. We would then able to evaluate the model by comparing the rolling averages of SUNFLUX/CS with the rolling averages for the event frequencies. There are still notable differences between the new predictive model (i.e. using SUNFLUX/CS) and the observed frequency (rolling averages applied). However, there is a noticable improvement - the predictability has improved. In the first submission, the coefficient of determination was approximately, 0.54, when the occurrence of the region of interest (ROI) was compared with the event frequency (as defined by the Nano-ranking approach) (i.e., an

OLS on Figure 9 from the first submission).

**Winter of 2022/23:**

The days during winter 2022/23 were cleaned again and one significant event was removed from the analysis because it was deemed to be a windblown event (see Fig.5 in this response). As a result of the additional cleaning, the new curve for the Aliaga method shows less non-zero frequencies. It is quite tricky to distinguish sometimes between windblown and events as was the case for Figure 5.

**Disagreement with the predictive model**

The model was altered and instead of using the part of the back trajectory that traverses through the mixed-layer, we used the entirety of the back trajectory. By doing this the agreement improved for July 2023. See the difference in the two plots below, particularly in relation to July 2023:

Figure 2: The sun/cs where only the parts of the back-trajectory which traversed through the mixed-layer were used.

We used the air mass origin and rainfall to explore the reason for the disagreement between observation and predictive parameter (solar insolation/CS). However, we came to the conclusion that the reason for the disagreement in the first submission (i.e., Fig. 9 in the first submission) was because of how we accumulated the solar radiation parameter along the back trajectories. We can see it when comparing Figures 2 with 3. The reason for the previous disagreement was only taking into account the endpoints of back trajectories which passed through the mixed-layer (ML).

Figure 3, we tried to explore when the new model (i.e., SUNFLUX/CS using the entire back trajectory) showed the greatest disagreement. We looked at meteorology, Air Mass Origin, Biological activity, Sea-ice regions, and Wet Removal (i.e., accumulated precipitation). However, we could not find anything conclusive.

**Interpolatied timeseries for the likelihood of NPF occurrence (PNF, D)**

A new approach was taken to plot the timeseries of the ROI, essentially the hourly data points that correspond to the ROI are given a binary value of 1 and those outside are given a value of 0. Then, the timeseries is transformed into daily averages (previously hourly) before a rolling mean is applied. So now, we have done away with using weekly averages in the process.

We have also compared the new P(NPF,D) with both the Nano-ranking and the DalMaso frequencies. However, to do we needed to apply some degree of smoothing - as they were all in binary i.e., 1s and 0s.

**First time the Nano-ranking method is used in the Arctic**

Figure 3: The replaced 30-day rolling mean plot (above, first one) taking the accumulated solar insolation from the entire back trajectory (BT) (all 6 hours prior to arrival, as opposed to just endpoints within the mixed-layer). the additional plots below show the solar insolation for 6 hours (entire BT), ML BT, the accumulated rainfall (entire BT) and the condensation sink (black line). The lower plot shows where the air masses came from in regard to the prevailing basin/origin.

We provide some general comments on the use of the Nano-ranking method for the Arctic, as special care needs to be taken when dealing with windblown snow events and in-cloud events (atop mountains).

We write the following:

This study is the first to utilise the Nano-ranking method in the Arctic. It should be stated that special care needs to be taken in treating wind-blown events separately from NPF events. Furthermore, this study also highlights potential problems when sampling lines are submerged in dense clouds. It is recommended that relative humidity, visibility and/or web cam footage is used to help screen for these in-cloud events when sampling at high altitudes.

**Rainfall:**

Rainfall is an important parameter when discussing NPF. The tricky aspect is that not only does NPF need rainfall (i.e. to reduce CS) but it also needs the absence of rainfall (i.e. clear skies and solar radiation). The exact time in which rainfall occurs is critical to whether or not an NPF event will take place. The presence of clouds could be a better predictor of removal over rainfall. Due to difficulties predicting rainfall in the Arctic. The exact nature of the rainfall will also impact i.e. its intensity (e.g. drizzle') will also impact its ability to reduce CS.

**Minor concerns:**

Considering solar radiation along the back trajectory is an excellent idea as this would give

a better indication of photochemical activity compared to in situ solar radiation and allows for appropriate time for oxidation of precursor gases to occur although I am now wondering if the authors considered something similar for the CS as well. Obviously, HYSPLIT cannot give an indication of the CS along the back trajectory but have the authors considered lagging the CS or possibly using accumulated precipitation along the back trajectory? They demonstrate precipitation and cloud processing has a large effect on NPF in Fig 11 and state this on 683. Tunved et al. (2013) and Khadir et al. (2023) also demonstrate the effect of accumulated precipitation on ultrafine aerosol particles nicely from ZEP. The authors demonstrate the NPF events will increase the CS, so I am wondering if the authors explored other variables, such as accumulated precipitation, to include in the simplified predictive model.

For the simplified predictive model, how did the authors calculate the line defining the ROI? Was it based on visual inspection or adjusted based on a criterion? I am very curious as to how did the authors calculate these averages "mean (median) 66% (68.4%)"? Did they simply take the mean and median of the ratios in the ROI? The large differences in the number of observations in different regions of the ROI could make such averaging give misleading results. Were the ratios weighed by the number of observations in each grid cell, i.e., weighted mean or median? I am wondering if a weighted average or calculating the sum the numerators and denominators of each grid cell in the ROI (i.e., integrating over the ROI) would be a more robust statistic. Could the authors please clarify this?

The ROI does have a higher likelihood of observing an NPF day compared to outside. However, there are grid cells with high likelihoods outside of the ROI although these grid cells do have a small sample size. I am wondering if the authors could comment on these false negatives in their model.

**Response to accumulated precipitation comment:**

A lot of work was done to try and explore the connection between precipitation and NPF. However, the authors found that it was quite a complicated parameter to deal with in the context of NPF. Not only are there multiple parameters which could be used i.e., accumulated precipitation, rainfall event frequency, and intensity, but the timing of the precipitation is also key to understanding its effects. If we take accumulated precipitation, we would then have to decide at what point does precipitation downwind have little to no effect on the on-site NPF observations.

We explored the correlation between precipitation at various stages along the back trajectory and NPF activity (see Fig. 4). We also took various accumulation periods and compared it to the NPF activity and strength, however we only saw weak correlations. However, there is a strong link between accumulated precipitation and CS, i.e., negative correlation.

Here, in figure 4 I tried to correlate the rainfall at varies stages/time steps along the back trajectory to see if there was a strong correlation at a particular time step - we know we generally need clear skies but also we need a mechanism that removes the accumulation mode particles (i.e., lowering the CS). However, there was no good correlation found between certain time steps and the NPF event frequency. The point-biserial correlation coefficient was used as the NPF event frequency was in binaray form i.e., 1s (events) and 0s (non-events).

For the idea of "lagging the CS", a rolling mean of previous CS measurements could be used, but might add a layer of uncertainty.

**Response to ROI comment:**

The ROI was defined simply by visual inspection. The averages where simply calculated by a mean of the grid cell probabilities/likelihoods. No weighting was applied. We describe the measure-

Figure 4: The point-biserial correlation coefficient comparing rainfall at a certain time step with the NPF frequency.

ments inside the ROI as fairly rare events and therefore by their nature they offer few data points. I disagree that the averaging is misleading - it is what it is - they are few but when they occur an NPF event is likely to occur on that day.

Essentially, we are counting all the NPF data in the ROI and dividing it by all the data in the ROI. Is that what you meant by "integrating over the ROI".

We imposed a threshold of 1 i.e. grid cells had to have more than 1 data point to be valid - this removed a lot of these false negatives. What the other false negatives are linked to is unsure - however, it could be related to large NPF events that grow large enought to influence the CS significantly.

**Line by line comments:**

Line 49: Maybe it is worth mentioning ZEP is a mountain site when describing the geographic location.

Done. edge of Svalbard atop a mountain.

Line 55: "which matches the frequency" Here would be a good place to include the performance evaluation for their model.

Done. We show that the ratio of solar insolation (SI) over CS corresponds well to the frequency of NPF events ( $\mathbb{R}^2$  of 0.78).

Line 57: I would say the authors nicely demonstrate the polar night NPF events arrive from higher altitudes (Fig. S17) therefore they are not speculating.

removed the word speculate.

Lines 61-63: The two parts of this sentence are redundant. Please make this sentence more concise.

Changed to the following:

We also show that NPF events lead to an increase in the number of Aitken mode particles,

indicating that a significant proportion of the Aitken mode particles originate from NPF.

Line 63: "over 50 NPF events" Would this be more accurately expressed as a percentage of total NPF events for context? This would help contextualize the importance of NPF's contribution to CCN in the Arctic.

Agreed. Converted to a percentage by dividing by the total number of events i.e., 157 which is 37%. It reads as such Of the measured NPF events, 37% of them exhibited nucleation mode particles that grew beyond 25nm, a diameter representing the minimum activation diameter for particles to act as cloud condensation nuclei.

Line 90-92: Could the authors include more original references for this statement? Also, Carslaw, 2022 is not in the reference list.

Agreed. Added a review paper here - Li, J., Carlson, B.E., Yung, Y.L. et al. Scattering and absorbing aerosols in the climate system. Nat Rev Earth Environ 3, 363–379 (2022). https://doi.org/10.1038/s43017-022-00296-7. It was Carslaw, K.S. ed., 2022. Aerosols and climate. Elsevier. Taken from chapter 2.4 Aerosol radiative forcing

Lines 109-111: The authors mention both negative and positive ions but only give a reference for negative ions, can the authors include a reference for positive ions?

Added a reference to Ion-induced sulfuric acid-ammonia nucleation - T. Jokinen, M. Sipilä, J. Kontkanen, V. Vakkari, P. Tisler, E.-M. Duplissy, H. Junninen, J. Kangasluoma, H. E. Manninen, T. Petäjä, M. Kulmala, D. R. Worsnop, J. Kirkby, A. Virkkula, V.-M. Kerminen, Ion-induced sulfuric acid-ammonia nucleation drives particle formation incoastal Antarctica. Sci. Adv. 4, eaat9744 (2018). Line 122: Villum Research Station is the preferred name for this station.

**Corrected**

Line 125: The town of Barrow, Alaska was renamed to its original, indigenous name of Utqiagʻvik in 2016. However, the NOAA observatory located outside of the town is still referred to as the Barrow Atmospheric Baseline Observatory (BRW). I would encourage the authors to either refer to the town by its proper name or be explicit when referring to the atmospheric observatory.

**Changed to Utqiagvik for all mentions of Barrow**

Lines 128-131: Beck et al. (2021) also showed different species contributing to NPF and growth during different times of the year at these two locations, iodic acid at Villum during springtime and sulfuric acid/ammonium during summer; sulfuric acid/ammonium at ZEP during springtime and highly oxygenated molecules during summer. Could the authors please mention these important results from Beck et al. (2021)?

**Added. Reads like the following now:**

Beck et al. (2021) through a comparison of measurements from Ny-Ålesund and Villum, showed that different nucleation mechanisms can also occur at different times of the year at these two locations; at Villum iodic acid was found to be the primary driver of NPF events during springtime and sulphuric acid/ammonium during summer. At Ny-Ålesund, NPF was driven by sulphuric acid

(SA) and ammonia during the springtime and highly oxygenated molecules during the summer.

Line 132: Please define DMS and mention its marine, biogenic origins. The authors mention phytoplankton blooms on lines 86-87, here would be a good place to first mention DMS and its effect on NPF.

Small definition added as follows:

Instead, Dimethyl Sulphide (DMS), produced by phytoplankton, can serve as a precursor gas contributing to NPF.

Lines 132-135: As currently constructed, this sentence implies the correlation between DMS and chlorophyll a is the reason that DMS is an important source of nucleating vapors. While none of the information in this sentence is incorrect, the way the sentence is structured is misleading, please rephrase.

Changed to the following: Marine regions with strong chlorophyll a concentrations and DMS production capacity have been linked to increased concentrations of nanoparticles, which suggests that marine biogenic sources are an important source of nucleating and condensing material (Lee et al., 2020).

Line 146: "higher end of the accumulation mode" do the authors mean Aitken mode here?

Sentence cut into two as previous one too long. Changed to the following:

In this study, we present two and a half years' worth of measurements of particle and ion number size distributions, including three summer periods (in total data covers the period April 2022 – October 2024). Continuous measurements of sub-3nm up to 850nm particles were performed at the ZEP, Svalbard, providing unique information of the initial stages of nucleation and the subsequent growth.

Line 176: Are two CPCs in this sentence referring to the ones on the twin DMPS system? Or the two CPCs references in the next sentence? If referring to the DMPS system, then saying "total aerosol oncentration" for size segregated monodispersed aerosol is confusing.

Agreed. Rewrote the sentence and removed the word 'total'. The sentence now reads as follows: The aerosol particles were counted using condensational particle counters (CPCs) at a frequency of 1 Hz with the two CPCs. Particles 3 nm were measured using the Ultrafine CPC (UCPC) TSI model 3776 (cut-off of 2.5nm) coupled to the DMPS-1. Particles 10 nm were measured with the CPC TSI model 3010 (cut-off of 10nm) coupled to the DMPS-2.

Line 244: Would it be possible to show a figure of exemplary days for Class Ia, Ib, and II. NPF event in the supplement? Given there is a degree of subjectivity in the Dal Maso method, this would help readers who are unfamiliar with the method gain context for each event.

Agreed. Yes this would be possible. In supplement there are now examples displayed in Figure S12. The event types are also averaged and plotted and displayed in Figure S13 in the supplement.

Line 283: Was the growth rate calculated using the timestamp for the maximum rate of change for each size bin as listed in the text or the maximum concentration as listed on line 301? Equation

**1: Ni-j is not defined.**

The Growth Rate (GR) was calculated by finding the datetime which corresponds to the maximum rate of change for each size bin. Janne Lampilahti's npf-event-analyzer was used to calculated GR and uses a different method. Added a claffying sentence:

When using the npf-event-analyzer the max concentration method was used for the estimations of GR and J.

Well spotted.  $N_{i-j}$  is now defined.

Line 300-301: The link does not work and there is no repository named npf event analyzer. on jlpl's Github page.

Unfortunately, Janne Lampilahti's npf-event-analyzer is being currently worked on - it is not at the moment publicly available. The link has been removed.

Equation 3: CS is not in Eq. 3 but is listed in the description. Is it part of another term in the equation?

Removed. Not needed any more - related to a previous draft. CS was there because there was a previous additional equation that in the end was not used

Line 320: Was the starting height above ground level or mean sea level?

HYPSPLIT model is ran with kmsl=0 in the SETUP.CFG file which indicates heights are specified as Above Ground Level (AGL). Height is entered as meters AGL unless the mean-sea-level flag has been set. Text changed to the following:

Ensemble back trajectories were initialised every hour starting at the latitude and longitude of ZEP, and at a height of 250m above ground level (AGL).

Sects. 2.4: I commend the authors on calculating ensemble trajectories as this gives a more robust estimate of air mass source regions compared to a single back trajectory. They are also a bit more complex to work with. I am curious if the authors combined all ensemble members or used individual ensemble members for the calculation of chlorophyll a exposure, nucleation site estimation, and air mass source regions. Could the authors please explain this in the text?

When back trajectories were used in the study they were used as ensembles i.e., no averaging was performed to replace the ensembles to single back trajectories - that would certainly defeat the point. Added a clarifying sentence as follows:

No averaging of the back trajectories was performed, reducing them to single back trajectories.

Sect. 2.4.2: Could the equation for calculating chlorophyll a exposure be listed here? Readers might not be familiar with the work of Park et al. (2018). I am also curious as to why the authors calculated chlorophyll a exposure for 2.5 years of trajectories (no easy task) and do not show any of the results even though they are mentioned several times throughout the text.

This has now been added to the text.

Sect. 3.1.1: "Occurrence" and "frequency" are used interchangeably, I am wondering if it is better to be consistent to avoid confusion.

The word frequency appears 29 times and occurrence 22 times. I hear your point but I believe that the read will have no trouble understanding what is meant.

Lines 379-383: The authors do an excellent job of describing previous studies on NPF activity at ZEP, could the authors include studies from other Arctic sites to provide context.

**Added the line:**

Dome-C, in Antarctica and Alert, on Greenland are two other polar sites with annual NPF frequencies between 10-20% (Nieminen et al., 2018), comparable to ZEP.

Line 390: "maximum daily increase" should this be "maximum daily concentration" as listed on line 266?

**Correct. Changed**

Lines 400-401 and 407-408: This is where some quantitative statistical metrics could be included.

Correct. We have explored this since - calculating coefficients of determinations  $(R^2)$ , varying the averaging window and the number of hours of accumulated solar radiation. This was done for the event frequency and event intensity.

We notice that for intensity a much better relationship is observed when selecting for air masses within the mixed layer (see Fig. S27 in supplement,  $R^2$  0.5 when window is greater than 10 days). This is also true when SUN/CS is used. However, for frequency selecting all the altitudes produces slightly improved  $R^2$ s.

The original idea was that by selecting the air masses which traverse only through the mixedlayer, we potentially focus more on the influence surface emissions can have on NPF activity.

Line 411-412: What other precursor candidates are there for sulfuric acid production other than SO2?

**Added/changed the following sentence to:**

The solar intensity is a controlling factor in the production of SA (i.e. H2SO4) via the oxidation of SO2. In the Arctic DMS acts a significant source of sulphate, however other sulphur compounds can be oxidised e.g., carbon disulphide and hydrogen sulphide (Pei et al., 2021).

Lines 412-413: The CS is for condensing vapors and the CoagS is for coagulating particles, I think these terms are switched in these lines.

**Correct. It is written as follows:**

The coagulation sink (CoagS) acts as a sink for the newly-formed particles, and the CS acts a sink for nucleating vapours, thus high CoagS and CS can inhibit NPF.

Lines 417-418: Can the authors simply list the changes in cloud cover or their frequencies from Maturilli and Ebell (2018)?

The skewed pattern in both the solar insolation and NPF intensity ( $\Delta$ Nmax 2.8-5) can perhaps be explained by annual changes in cloud cover (see Maturilli and Ebell (2018) for frequency occurrence of cloudy sky, increasing to 80% during summer from an April minimum of 40-60%).

Lines 423-427: It is interesting to note the almost anti-correlation between DMS and  $\Delta N$ max 2.82-5.

Yes. DMS does not act as a indicator of NPF events it seems. We explored this further looking at some simple correlations however it is a very weak negative correlation.

Lines 434-436: Would the polar night events be better presented as a table? What is the meaning of 3ab and x2 in these lines?

Table added with extra information including duration and GR

Lines 438-439: The latitudinal dependency is explained but was there any longitudinal dependency of these polar night events?

It is not necessarily a dependency, but simply an observation. Eventally, all back trajectories end up going further south than 79°N.

Lines 439-441: Wouldn't the low temperatures experienced by high altitude air masses during this time of year also lower the volatility of any nucleating vapors, making the process more efficient?

Not sure.

Line 441: Utqiagvik is misspelled.

Spelled correctly.

Line 465: Could the authors mention the differences in the definition of the onset of nucleation between these studies?

I have read the Lee et al., (2020) paper again and it is unclear how they define the length of the NPF event. There are three types that they mention type I, II and undefined. I can only infer the definition. Type I is defined as  $N_{3-25}$  increasing significantly with subsequent particle growth, therefore the time in which the rate of change of  $N_{3-25}$  is greater than 0 should be used as the definition. For type II, this would be the same but without significant particle growth i.e. no change in the larger  $N_{25-60}$  particles. With 2 hours as the cut-off. Undefined sees an increase in  $N_{3-25}$  but for less than 2 hours. Hence, I would say that they define the length as the total time in a day that the rate of change of  $N_{3-25}$  is greater than 0. Added to this there are no lengths that are greater than a day (24-hours), whereas we use the entirety of the event - can be longer than a day. Hence, our lengths are longer. Furthermore, we judge the length of the NPF event by eye so we don't impose any diameter thresholds i.e., 3 nm 25 nm.

Sect. 3.1.5: The section is titled "Seasonality of Growth Rates and formation rates" yet no seasonality of formation rates is presented. I would suggest adding a subpanel to Fig. 4 which shows the seasonality of formation rates and mentioning the seasonality in the text. There should be a space between nm and hr-1, also in the figure axis label. What sites were included in Kerminen et al. (2018) for the Arctic?

For Kerminen et al. (2018), the polar sites include measurements taken on or at Canadian research icebeaker, research aircraft Polar 6, Barrow Observatory, Swedish icebraker Oden, and the high Arctic site Vllum research. You can see the list of polar sites in the following link.

https://iopscience.iop.org/article/10.1088/1748-9326/aadf3c/data

Formation rates.

The seasonality of formation rates has now been done

Sect. 3.1.6: I am curious why the authors did not compare Cv to the results from Beck et al. (2021) as well. Observations from MOSAiC are highly valuable in this context but Beck et al. was from Gruvebadet at the base of Mount Zeppelin. The MOSAiC year was also shown to be anomalous in terms of the Arctic Oscillation and transport from Eurasia during the winter months (Boyer et al., 2023). The caption of Fig. 5 reads as if the measurements from Boyer et al. cover the years 2022-2024.

This is a very good point, we have now included the data from Beck et al., (2021) and effectively replaced the Boyer et al., (2023). The data from Beck et al., (2021) matches really well to the estimated Cv. Some mention of the Boyer et al., (2023) is done because we find it interesting that it implies a latitudal gradient to Cv.

When we have mentioned Boyer et al., (2023) though we have added the following wording: The estimated Cv measurements cover the years 2022 to 2024, whilst the MOSAiC Cvap measurements cover one year from 2019-2020.

Lines 510-511: Could the other species possibly be HOMs? Which were reported by Beck et al. (2021) but not Boyer et al. (2024).

This could be the case given that Beck et al., (2021) also report higher HOMs towards of the summer where the disagreement between measured values and estimated is the largest. Now we have used Beck et al., (2021) instead of Boyer et al., (2023) the role of HOMs becomes a lot clearer.

Header of Table 1: The unit for CS needs a superscript. The unit for solar insolation is given as hW m-2 elsewhere, also should there not be a space between hW and m-2?

Made the superscript on CS. Changed to "hWm-2"

Caption and label of Fig. 7: I am wondering the term "Nucleation site" is confusing and would it be better to express it more generally as "Count per grid box" as expressed in previous publications (Heslin-Rees et al., 2024). Is the NAIS defining the nucleation sites or is it HYSPLIT?

Nucleation site refers to an estimation of where the nucleation took place. For example, if we are measuring particles 4 hours into an event then we assume nucleation took place 4 hours ago. We initialise the trajectory backwards for 4 hours.

Lines 575-578: I would encourage the authors to include their results here as this is an interesting finding that chlorophyll a exposure is not correlated to DMS.

For sure, I can put it in the supplement in a whole subsection S12

Lines 578-583: Can the authors give examples of "various other meteorological and environmental factors"? Also, could there any methodological factors influencing the lack of correlation

between chlorophyll a exposure and DMS, such as inaccuracies in HYSPLIT and satellite retrievals of chlorophyll a due to sea ice and cloud cover?

It is important to note that we do not know the exact reason for the lack of a correlation, but of course it could be related to inaccuracies in HYSPLIT and satellite retrievals of chlorophyll a as a result of sea ice and cloud cover. I have altered the text to include these as other possible explanations

Lines 585-588: I would suggest removing this sentence as it does not add to the discussion nor explain their results, although this is only my opinion.

Yep, agreed. This sentence is confusing to understand and does not add anything concrete. It has been removed.

Lines 589-603: The authors do a nice job of explaining the differences between the Greenland Sea and the Barents Sea for NPF source regions, although another region highlighted in Fig. 7 is the Arctic Ocean. Could emissions from sea ice, e.g. iodine and sympagic algae be contributing to this observation? This might be worth mentioning.

**Added a sentence discussing the Arctic Ocean.**

Lines 606-608: Here is an example of where quantitative statistics could be used to support the results. For instance, the distributions of solar insolation on NPF vs non-NPF event days could be compared or the anomaly of the CS to the seasonal average could be analyzed for NPF event days. Such an analysis would make their justification of the simplified predictive model more robust and convincing.

Yep, agreed. Work was done to analyse whether there was a statistical difference between the frequency and intensity parameter for NPF and non-NPF days. The solar insolation parameter was used to differentiate between the two categories. The Mann/Whiteney U test (also known as the Wilcoxon rank-sum test) was used to test various hypotheses related to the NPF frequency and NPF intensity.

- H0: distributions of solar insolation and/or CS for NPF events and non-event are identical
- H1: distributions are not identical

Lines 613-622: The units for solar insolation are sometimes given as hWm-2 and sometimes as Wm-2, additionally there should be a space between W and m-2 otherwise it reads as if both units are in the denominator. Units for CS are missing.

Agreed this was altered in the text where found.

Line 617: I am curious how this relationship is for the Dal Maso method, does it still hold? Fig 8: I would suggest making the two subpanels the same size and please use a colorblind friendly color bar.

We have added Fig 12 to the main manuscript which compares the R2 for both classifications for a comparsion of SUNFLUX/CS and different event frequencies.

Colour bar changed to one that is colourblind appropriate

Lines 627: Could the authors quantify how much less likely NPF events are outside of the ROI?

Yep. We have done so. The following sentence has now been added:

In addition, the ROI and non-ROI represented 23% and 77% of the data respectively; for the ROI 62.5% of the hourly data were linked to NPF days, as opposed to 23.8% in the non-ROI.

Line 693: Another example of where quantitative statistics would help as this "substantial predictive value" is not demonstrated.

Yep. A whole new figure has been added detailing the difference between the ROI and non-ROI in terms of probability

Line 701-703: Figure 13 is not referenced in the text. Should it be referenced in this sentence?

Reference to Figure 13 is added to the text after the following sentence: During NPF events, including formation and subsequent growth the total number concentration of Aitken mode particles (5-70nm) is 3.6 times the amount compared with pre-event conditions (see Fig. 13).

Line 709: Why not be precise with an exact number of NPF events that grew over 25nm?

Okay. Used the exact number.

Line 710: In the Arctic summertime, the lack of accumulation mode particles can drive the supersaturation to high values, activating small particles as the authors detail nicely, thus 25 nm is a lower limit. Could the authors give the number of NPF events that surpass other thresholds for activation, such as 60, 70, or 80 nm (the exact thresholds are left up to the authors). This would give a range of the number of NPF events that could contribute to CCN.

Agreed. A short sentence is added to the text and reference the number of particles observed which reach the various sizes 60, 70 and 80nm.

Line 712-713: Figure 12a is not referenced in the text, should it be referenced in this sentence? What instruments were used to calculate the mean and median diameter and how were these calculated? Using the mean and median, it appears that the diameter reaches 25 nm after approx. 30 hours. The shading on the lines (which is not mentioned in the figure caption) appears to reach 25nm after 20 hours.

Agreed. Added the reference to Fig. 12a.

Line 753: This accuracy is not quantified.

Agreed. We have tried to analyse the reliability of our model using a simple ordinary least square (OLS). We present these results in the supplement but refer to them in the text.

**Supplement:**

Fig. S1 caption: Could the authors list the names of all the parameters in paratheses? The authors state 5 tubing parameters but more than 5 are given.

I just re-wrote it to say "sets of measurements" as it five 5 tubes with length and diameter measurements stated.

Fig. S2a: The legend significantly overlaps with the data and the abbreviations in the legend are not given.

Added more information. Cloudnet data where Clear Sky (CS), Drizzle (DR), Ice (I), Melting Ice (MI), Aerosol (A), Aerosol and Insects (A+Ins), yellow line is the cloud base height (cbh), Cloud Droplet (CD) is purple, Drizzle and cloud droplet (DR+CD), Ice and Supercooled droplets (I+SCD), Melting Ice and Cloud Droplets (MI+CD), Insects (I), red line is the cloud top height (cth) and ZEP is the height of the observatory at 474m a.s.l. into the caption.

Fig. S6 caption: What is the meaning of "These are overlapping – improvement representation"?

There is no meaning the caption is just wrong. It now reads:

"Number of hours of data (i.e. time steps) containing observed windblown snow periods (black), cloud-events when the station is submerged within dense clouds (dark blue), and when both events are occurring at the same time. Note the increased frequency of windblown snow events ('snow', black bar) in the winter and early spring and the increased frequency of In-cloud events ('cloud', blue bars) in the summer months."

S2.5: The authors state "There are numerous papers which highlight this common discrepancy." but none are cited, please cites numerous papers to support this statement.

```
Added "common discrepancy e.g., Dada et al. (2023)." S2.5 below Fig. S9: Fig. X? Should this be Fig. S11? Yep, wrote "see Fig. S11"
```

Fig. S10: The caption does not explain the figure at all. Please rewrite the caption to accurately describe the figure. Maybe this text was meant to be included elsewhere? Y axis of Figs. S10 and S11: What is dndlDp? Should this be dN/dlogDp as used elsewhere in the manuscript? From the caption, I gathered this is the slope of the regression line, but "Coef." is listed in the label. Finally, "(DMPS/NAIS-5)" reads as if 5 was subtracted from the ratio of the DMPS to the NAIS, although it was clear from Sect. 2.2 that it is NAIS5. Please adjust the y axis label to be clearer.

The correlation of determinations and gradients when comparing temporally collocated DMPS and NAIS-5 particle number concentrations. The data is re-gridded such that direct comparisons can made for each size range. Each comparison for a respective size bin takes the form of an ordinary least squares (OLS) regression such that the slope is displayed on the y-axis and the coefficient of determination (R2) is displayed by the colour of the circle marker. Red presents a positive correlation whilst blue a negative correlation. Dada et al., (2023) found that the NAIS overestimated the particle concentrations by a factor of 5 in the overlapping size range of 6–42 nm, and applied a correction factor of 1/5 to the NAIS measurements. Hence, a blue horizontal line represents the factor used by Dada et al., (2023). The red line signifies the average slope across all the different size bins (i.e., diameters).

Fig. S12: There is no (b) panel in the figure. Is "N int max" in the title supposed to indicate

 $\Delta$ Nmax, 2.8-5?

Changed.

Fig. S13: What is dNint,max and d\_N\_int\_max? These terms are not defined. Should the ratio for the mapping of g1, g2, and g3 be 1:1:1 since there are three groups but only two numbers in the ratio?

Plot changed and caption improved. Fitting for the intensity parameter. The  $\log(\Delta N2.8\text{-}5nm)$  distribution is depicted. The three different intensity groupings i.e., g1, g2, g3 are created by taking the log of the intensity parameter i.e., the daily maximum in the concentration of particles between 2.8 and 5nm ( $\Delta N2.8\text{-}5nm$ ). The fitting of the groups is based on the log of the strength parameter. A visual inspection and trial and error is used to fit the Gaussian curves. Note that the mapping used between  $\log(\Delta N2.8\text{-}5nm)$  and the class of event i.e. g1, g2 g3 is 1:1. The dashed lines are calculated by calculating the distance between the adjacent Gaussian curve curve centres (centre c1 for curve g1, centre c2 for curve g2, centre c3 for curve g3). See Section about NPF mode fitting (Sect. 2.1.2) in Aliaga et al. (2023).

Fig. S16: Should the units for the y label be in nm? Also, no color bar is given.

Units changed and colourbar added. New figure.

Fig. S17: What are the vertical dashed lines in this figure?

More information was placed in the caption to explain the dashed lined. The dashed lines represent the average altitudes for the various subsets of the data set.

Figs. S2, S7, S8, S16 present a lot of information, therefore I would suggest making these figures larger, possibly the width of the page. This would make it easier for the reader to grasp all the details presented in these nice figures.

Increased the size of these figures to fill the entire width.

A bibliography is missing from the supplement. Bibliography has been added to the supplement.

**2 Reviewer 2: Anonymous Referee #2, 14 Apr 2025**

**General Comments**

The study "Drivers governing the seasonality of new particle formation in the Arctic" by Heslin-Rees et al. presents an extensive dataset of particle and ion measurements in the Arctic atmosphere.

With statistical analysis of the dataset, the authors characterize the frequency of new particle formation and try to identify the circumstances of its appearance at the Zeppelin station on Svalbard. Further, they analysed the air mass origin and linked it to the likelihood of NPF events.

I would recommend the study to be published after addressing my minor concerns.

Main Manuscript

Specific comments (Numbers refer to Line numbers in the original manuscript PDFs)

Is it possible to quantify how often ZEP was inside the boundary layer and how often it was above? Line 463 and following: Might the boundary layer height have an impact on the NPF timing and in general on the frequency as well?

I would say that it is quite different to estimate the time ZEP spends within the boundary layer. Such a calculation is beyond the scope this paper. Calculating the time spent within the boundary layer is difficult given the topography and mountain terrain. The exact nature of the evolution of the boundary layer is not needed, we simply wanted to highlight that over the course of the day precursor vapours can and do accumulate.

Changed to a simpler statement:

"This could reflect that during the NPF season there is a delay of approximately 8 hours from the minimum solar intensity to the onset of NPF, and this could be related to the time it takes for precursor gases to accumulate"

The GR during July 2024 (if I read the month from the x-label correctly) is considerably higher than during other months. Do the authors have an idea why this might be the case?

We looked into the GRs again to check. More days have been cleaned further. There was an event where the GR was to unreliable to calculate i.e., 2024-07-18 which has since been removed from the calculations. The bar plot has been updated.

I believe I do not quite understand the two values for the rolling mean - e.g. in Fig. 9 the authors show the 30 day rolling mean - what does the min. window 7 d mean exactly? Are periods with less than 7 days excluded?

The rolling average works by having a window of X days (e.g. 30 days) and a minimum requirement that the window has to be of a length of X days (i.e. in this case 7 days long). In Python, the minimum window is defined as "Minimum number of observations in window required to have a value; otherwise, result is np.nan"

The rolling windows and minimum values have since been changed.

This figure has since changed. The figure takes the daily timeseries for the observations (Dal Maso and Nanoranking classification schemes) and the timeseries of the occurrence of ROI values. The ROI occurrence was converted to daily averages before a rolling mean was applied.

I would recommend rounding the delta N boundary of 2.82 to 2.8 as the accuracy of the NAIS, to my knowledge, does not capture 1e-2 nm.

Yes, we have changed the number points from 2.82 to 2.8.

The value essentially comes from the fact that after using the nais-processor, we get Dps as follows:

and the diameters in the processed data are as follows, in nm: ... 2.233, 2.505, 2.812, 3.155...

for the intensity parameter we explored using various size intervals and decided upon using the size bins larger than the processed "2.812" nm size bin.

In the NAIS-5 manual the diameter ranges for electrometer 7 & 8 are 1.84- $2.58\,\mathrm{nm}$  & 2.14- $2.91\,\mathrm{nm}$ . Hence we would be on the upper end of electrometer 8.

For the inverter which is used i.e. the "particles-neg-v14-hrnd-elm25-chv.inverter" and "particles\_pos\_v14\_hrnd\_elm25\_chv" then the size ranges for different inverters are for the electrometer "7" are 2.005-3.174nm, and for Elm. "8": 2.353-3.883nm.

Hence, there is some uncertainty as to the actual minimum size of the intensity parameter. It remains unclear what particles have been chosen to be included in the intensity parameter i.e. "2.8-5nm".

The authors use the vapor concentrations contributing to NPF in chapter 3. Those should also be mentioned in the introduction, so that the readers are on the same page and understand why they are being part of this discussion. Generally, the contributing vapors for NPF in the Arctic atmosphere should be discussed a little more in detail even though the vapors were not measured here.

More mention of precursor vapours in the Arctic was placed in the introduction. It now reads like this:

Beck et al. (2021) through a comparison of measurements from both Ny-Ålesund and Villum, showed that different nucleation mechanisms can also occur at different times of the year at these two locations; at Villum iodic acid (IA, HIO3) was found to be the primary driver of NPF events during springtime, and sulphuric acid (SA, H2SO4) -ammonium (AM, NH4+) driven events were shown to occur in the summer. At Ny-Ålesund, NPF was driven by SA-AM during the springtime and highly oxygenated molecules (HOMs) during the summer. There is no evidence that anthropogenic precursors contribute to NPF at ZEP (Schmale and Baccarini, 2021). Instead, Dimethyl Sulphide (DMS), produced by phytoplankton, can serve as a precursor gas contributing to NPF. Marine regions with strong chlorophyll a concentrations and DMS production capacity have been linked to increased concentrations of nanoparticles, which suggests that marine biogenic sources are an important source of nucleating and condensing material (Lee et al., 2020).

we have mentioned sulphuric acid (H2SO4), ammonium (AM, NH4+), iodic acid (IA, HIO3), and DMS in the introduction.

line 337: DMS is not only a precursor to sulfuric acid, but also methanesulfonic acid.

added the following sentence:

It should be noted that DMS is not only a precursor to H2SO4, but also MSA.

line 416: This is more a question of curiosity: Do the authors think that dilution due to stronger insolation and thus elevated boundary layer heights during summer might play a role in the reduced NPF intensity?

I do not particularly understand this comment. Solar insolation leads to an increase in NPF intensity, not a decrease. The diluation that comes with the increase in solar insolation and the boundary layer height will play a role but we cannot qualify it without properly understanding the evolution of the boundary layer.

The dip in NPF event intensity corresponds with a dip in solar insolation and increases in CS.

line 500: I believe the sentence "HOM concentrations are very low during spring and increase in May, however it still shows their importance in contributing to Cv." needs a bit more elaboration. The abbreviation HOM is not yet mentioned or explained in the manuscript. Why do the authors think they are contributing to the Cv? Because the sum of IA, MSA and SA are not explaining the Cv alone?

HOMs is now defined earlier. Yes, essentially, we think that HOMs is playing a role because it has been should to play a role in Ny-Ålesund by Beck et al., (2021). Furthermore, HOMs nicely shows that when taken into account along with the other condensable vapours - it accounts for the estimated Cvs.

line 540: The theory that during autumn months the nucleation pathway is rather dominated by organics which might show in the cluster ion relation sounds plausible. However, I would also keep in mind that during these months the NPF frequency was also considerably lower and I am not sure how much weight one can give the 5, or 3 events, respectively, in October / November.

HOMs begins to dominate from June onwards. There are still plenty of events taking place (as seen from Fig. 1). The shift to positive chargers is continuous from negative in May to positive in September. We have noted that events in September, October and November are few. 2022 (4+2+1=7), 2023 (7+5+3=15) and 2024 (6+1=7). In total, 29 events occur after September.

We wrote:

"However, it needs to be noted that there are few events during September and beyond (29 in total)."

line 577: What are the correlations of 2022 and 2023 and how do they compare to Park et al. (2018)?

We have put the correlation plots for 2022 and 2023 in the supplement now.

line 696 and following: were there any CCN instruments on site as presented in Karlsson et al., 2021 which could be supporting the analysis and impact on CCN production via NPF in this paper?

Yes, there were CCN instrumentation on site. One set up at Zeppelin Observatory is based on the Ground-Based Counter Virtual Impactor, in which cloud droplets are sampled, dried and the interstitial part ("activated" aerosol) is measured. However, we wanted to avoid using such data in this study. There are papers we refer to which have used CCN data to study the link between CCN and NPF e.g., Kecorius et al., (2019).

Fig. 1: I recommend having more details in the figure caption to guide the reader to understand more quickly, which y-axis belongs to which parameters.

Agree. Changed it to the following: "Fig 1: Number of days per month classified as either Class Ia (red), Ib (orange), II (yellow), Undefined (green) and Non-event (grey). Left-hand axis is the

number of days classified based on the total number of valid measurement days. Right-hand side is the fraction of events (i.e. Class Ia, Ib, and II) over the total number of days (i.e. Class Ia, Ib, II, Undefined and Non-event). The fraction is given by Event/Total. Almost three years of Neutral cluster and Air Ion Spectrometer (NAIS) particle number size distribution data is classified using the Dal Maso et al. (2005) classification system."

Is it possible that 31 days were included in June 2023? From the bar plot it reads like the total number of days included are 31 as during July and August

Yes, well-spotted the day 2nd of June 2023 was classified as both "Class II" and "Undefined", hence there was an extra day as this was counted twice in the sum.

Fig. 2: I suggest using more distinct colors.

The bright yellow colour representing solar insolation accumulated within the back trajectories entire path is changed to 'gold'. In the end this was the solar insolation parameter that correlated best.

Fig. 4: I suggest adding a legend for the figure (mean, median) even though the box-whiskers plot makes them self explanatory. The x-axis label is a bit difficult to read, it is not fully clear which month belongs to which tick-mark.

Yep, I agree. A new legend has been added.

Fig. 8: Do I understand this correctly, that the colorbar (panel a) is the count of negative ions? Which sizes are shown here? Do the authors have an explanation for the high concentrations of ions during low solar flux and low CS, e.g. is it linked to snow or cloud events, or are these days excluded here?

Fig. 8 Panel a) refers to the number of data points with respective 6-hr accumulated sun flux and concentration sink values. Panel a) is not related to NPF. More work has now been done to explain this figure. Also, the figure has changed quite a bit since the first submission.

But essentially, what we are doing is just dividing two grids by one another. the first grid is data representing events and the second is all the data - this is why we call it a normalisation.

Therefore the value in the grid boxes after "normalisation" represent a likehliood.

Fig. S16: The y-axis label seems to be wrong, I believe it should read nm, not meter.

**Corrected to [nm]**

**Technical remarks:**

Line 110: "both" is repetitive

Corrected

Line 570: "at ZEP" is repetitive

Corrected

Line 720: typo: "can contribute"

Corrected

Line 751: typo: "rapid" Corrected, on line 753

**Additional changes**

I decided to remove the event during 2023-01-08 from the data being processes for the Nanoranking classification. It is thought that this event is in connection with the wind-blowing event taking place, however, in the in the particle-channel it appears the event has growing nucleation mode though. However, it is an event that might be worth further investigation as a wind-blow snow event that triggers the growth of a nulceation mode.

Figure 5: Removed for the purposes of the Nano-ranking classification and labelled as 'Undefined' according to the Dal Maso scheme.